

# Computationally efficient methods for large-scale atmospheric inverse modeling

Taewon Cho[1], Julianne Chung[2], Scot M. Miller[3], and Arvind K. Saibaba[4]

[1]Department of Mathematics, Virginia Tech, Blacksburg, VA
[2]Department of Mathematics and Computational Modeling and Data Analytics Division, Academy of Integrated Science, Virginia Tech, Blacksburg, VA
[3]Department of Environmental Health and Engineering, Johns Hopkins University, Baltimore, MD
[4]Department of Mathematics, North Carolina State University, Raleigh, NC

**Correspondence:** Julianne Chung (jmchung@vt.edu)

**Abstract.** Atmospheric inverse modeling describes the process of estimating greenhouse gas fluxes or air pollution emissions at the Earth's surface using observations of these gases collected in the atmosphere. The launch of new satellites, the expansion of surface observation networks, and a desire for more detailed maps of surface fluxes has yielded numerous computational and statistical challenges for standard inverse modeling frameworks that were often originally designed with much smaller data sets

in mind. In this article, we discuss computationally efficient methods for large-scale atmospheric inverse modeling and focus on addressing some of the main computational and practical challenges. We develop generalized hybrid projection methods, which are iterative methods for solving large-scale inverse problems, and specifically we focus on the case of estimating surface fluxes. These algorithms confer several advantages. They are *efficient*, in part because they converge quickly, they exploit efficient matrix-vector multiplications, and do not require inverting any matrices. These methods are also *robust* because they

can accurately reconstruct surface fluxes, they are *automatic* since regularization or covariance matrix parameters and stopping criteria can be determined as part of the iterative algorithm, and they are *flexible* because they can be paired with many different types of atmospheric models. We demonstrate the benefits of generalized hybrid methods with a case study from NASA's Orbiting Carbon Observatory 2 (OCO-2) satellite. We then address the more challenging problem of solving the inverse model when the mean of the surface fluxes is not known a priori; we do so by reformulating the problem, thereby extending the

applicability of hybrid projection methods to include hierarchical priors. We further show that by exploiting mathematical relations provided by the generalized hybrid method, we can efficiently calculate an approximate posterior variance, thereby providing uncertainty information.

## 1 Introduction

Numerous satellites and ground-based sensors observe greenhouse gas and air pollution mixing ratios in the atmosphere. A

20 primary goal of atmospheric inverse modeling (AIM) is to estimate emissions or fluxes at the Earth's surface using these observations (Brasseur and Jacob, 2017; Enting, 2002; Michalak et al., 2004; Tarantola, 2005).





The number of greenhouse gas and air pollution measurements has greatly expanded in the past decade, enabling investigations of surface fluxes across larger regions, longer time periods, and/or at finer spatial and temporal detail. Carbon dioxide ($CO_2$) offers an illustrative example. The Greenhouse Gas Observing Satellite (GOSAT), the first satellite dedicated to monitoring $CO_2$ from space, launched in 2009 and collects $\sim 1 \times 10^3$ high quality observations globally each day. NASA's Orbiting Carbon Observatory 2 (OCO-2 satellite) launched in 2014 and collects $\sim 100$ times more high quality observations, and upcoming satellites like the Geostationary Carbon Observatory (GeoCarb) could collect up to $\sim 1 \times 10^7$ each day (though some of these observations will likely be unusable due to cloud cover) (Buis, 2018; Crisp, 2015; Eldering et al., 2017; Nakajima et al., 2012). These new observations are complemented by an expanding ground-based network of observations (NOAA Global Monitoring Laboratory) and expanded aircraft observations, including partnerships with several airlines to measure atmospheric $CO_2$ from regular commercial flights (Machida et al., 01 Oct. 2008; Petzold et al., 2015).

This expanding network of atmospheric observations presents numerous computational challenges for estimating fluxes using inverse modeling and for quantifying uncertainties in the estimated fluxes. The design of new methods to improve the computational feasibility of large atmospheric inverse problems has been the focus of numerous recent publications, see e.g., (Baker et al., 2006; Bousserez and Henze, 2018; Chatterjee and Michalak, 2013; Chatterjee et al., 2012; Gourdji et al., 2012; Henze et al., 2007; Liu et al., 2020; Meirink et al., 2008; Miller et al., 2020, 2014; Yadav and Michalak, 2013; Zammit-Mangion et al., 2021). The computational challenges are many. First, large-scale inverse models based on Bayesian statistics often require formulating very large covariance matrices, calculating matrix-matrix products with those covariance matrices, and/or solving linear systems with those matrices. Second, existing inverse models often assume a Gaussian prior distribution for use with Bayes' theorem, where the prior mean vector and covariance matrix are required. Statistical approaches to estimating these covariance matrix parameters (e.g., restricted maximum likelihood estimation or Markov Chain Monte Carlo methods) are often difficult to implement for extremely large inverse problems (Ganesan et al., 2014; Michalak et al., 2005), and a common approach is to populate the covariance matrices using expert knowledge. Third, fluxes often need to be estimated using iterative optimization algorithms for very large problems, and convergence of these algorithms can be slow (Miller et al., 2020). Fourth, calculating uncertainties in the estimated fluxes can be computationally prohibitive.

**Overview of features and contributions.** The purpose of this study is to integrate several state-of-the-art computational and mathematical tools with AIM—tools that have been developed for and have had considerable success in other scientific fields (e.g., passive seismic tomography, medical imaging). Specifically, we investigate the use of generalized hybrid (genHyBR) projection methods for surface flux estimation and extend their use for inverse problems where the mean of the fluxes is not known *a priori* (sometimes referred to as geostatistical inverse modeling) (Chung and Saibaba, 2017; Saibaba et al., 2020). We address these challenges in our present work.

Building on prior work in (Miller et al., 2020), we propose a unified computational framework for large-scale AIM with the following features:

1. We describe iterative genHyBR methods that are computationally efficient since they typically converge in a few iterations, are efficient in terms of storage, and work for very large satellite-based inverse problems. For example, we





demonstrate the performance of genHyBR methods on two case studies previously considered in (Miller et al., 2020). In the larger case study, we solve an inverse problem with $9 \times 10^6$ unknown $CO_2$ fluxes and $1 \times 10^5$ $CO_2$ observations.

2. We extend these methods to handle the case where the mean of the prior distribution is unknown, making genHyBR applicable to a broader range of inverse modeling applications that are common in the atmospheric science community (e.g., geostatistical inverse modeling).

3. Our approach is flexible in that it can be combined with any atmospheric transport model (e.g., either Lagrangian, particle-following models or the adjoint of an Eulerian model), and it can be used with a wide variety of covariance matrices for the unknown parameters and the noise.

4. Our framework also allows for efficiently estimating regularization parameters as part of the reconstruction, thus making it easier to objectively estimate the hyperparameters or covariance matrix parameters as part of the inverse model. We focus on the discrepancy principle (DP), which requires prior knowledge of an estimate of the noise, but provide alternate methods such as the unbiased predictive risk estimator and the generalized cross validation, the latter of which does not require prior information regarding the noise level.

5. During the solution of the estimates, our solver stores information about the Krylov subspaces that can be used to estimate the posterior variance (at minimal computational cost), which gives insight into the uncertainty in the reconstructed solution. More precisely, evaluating uncertainties does not require additional model evaluations.

An overview of the paper is as follows. In Section 2, we describe the problem setup from a Bayesian perspective. In Section 3, we describe generalized hybrid methods for atmospheric inverse modeling. We show how to efficiently compute the maximum a posterior (MAP) estimate and uncertainty estimates (e.g., posterior variance). The focus of this section is on the fixed mean case. We briefly mention an extension to the unknown mean case, but defer most of the details to the Appendix B. Numerical results are provided in Section 4, and conclusions can be found in Section 5.

## 2 Bayesian approach to inverse modeling

An AIM will estimate greenhouse gas fluxes or air pollution emissions that match atmospheric observations, given an atmospheric transport model. It can be represented as an inverse problem of the form

$$z = \mathbf{H}s + \epsilon \tag{1}$$

where $z \in \mathbb{R}^m$ is a vector of atmospheric observations, $\mathbf{H} \in \mathbb{R}^{m \times n}$ represents the forward atmospheric transport model, $s \in \mathbb{R}^n$ is a vector of the unknown surface fluxes or emissions, and $\epsilon \in \mathbb{R}^m$ represents noise or errors, including errors in the atmospheric observations ($z$) and in the atmospheric transport model ($\mathbf{H}$). Note that we assume $\epsilon \sim \mathcal{N}(\mathbf{0}, \mathbf{R})$ where $\mathbf{R} \in \mathbb{R}^{m \times m}$ is a positive definite matrix whose inverse and square root are inexpensive (e.g., a diagonal matrix with positive diagonal





entries). The goal of the inverse problem is to estimate $s$ given $z$ and $\mathbf{H}$. The inverse problem may be ill-posed or under-constrained by available observations. Therefore, it is common to include prior information to mitigate the ill-posedness. We describe two different priors: fixed mean and unknown mean.

**Fixed mean.** A common approach in Bayesian inverse modeling is to model $s$ as a Gaussian random variable with a fixed, known mean $\boldsymbol{\mu} \in \mathbb{R}^n$ and prior covariance matrix $\lambda^{-2}\mathbf{Q}_s \in \mathbb{R}^{n \times n}$. In many cases, this known mean ($\boldsymbol{\mu}$) is an emissions inventory, a bottom-up flux model, or a process-based model of $CO_2$ fluxes (Brasseur and Jacob, 2017). This approach is also known in the literature as *Bayesian synthesis inversion*. Using this framework, the prior distribution of $s$ is given as follows:

$$s \sim \mathcal{N}(\boldsymbol{\mu}, \lambda^{-2}\mathbf{Q}_s). \tag{2}$$

We assume that matrix $\mathbf{Q}_s$ is defined by a covariance kernel that describes the spatial and temporal variance and covariance in the prior distribution (Rasmussen and Williams, 2006). Furthermore, $\lambda$ is a scaling parameter that is known *a priori* or has to be determined prior to the inversion process. The posterior distribution can be obtained by applying Bayes' theorem $\pi(s|z) \propto \pi(z|s)\pi(s)$, which takes the form

$$\pi(s|z) \propto \exp\left(-\frac{1}{2}\|\mathbf{H}s - z\|_{\mathbf{R}^{-1}}^2 - \frac{\lambda^2}{2}\|s - \boldsymbol{\mu}\|_{\mathbf{Q}_s^{-1}}^2\right), \tag{3}$$

where $\|x\|_{\mathbf{M}}^2 = x^\top \mathbf{M} x$ for any symmetric positive definite matrix $\mathbf{M}$, and '$\propto$' denotes a proportionality constant. The maximum a posterior (MAP) estimate corresponding to this posterior distribution can be obtained by solving the optimization problem

$$s_{\text{post}} := \underset{s \in \mathbb{R}^n}{\arg\min} \frac{1}{2}\|\mathbf{H}s - z\|_{\mathbf{R}^{-1}}^2 + \frac{\lambda^2}{2}\|s - \boldsymbol{\mu}\|_{\mathbf{Q}_s^{-1}}^2. \tag{4}$$

Alternatively, it can be computed by solving the system of equations

$$(\mathbf{H}^\top \mathbf{R}^{-1}\mathbf{H} + \lambda^2 \mathbf{Q}_s^{-1})s_{\text{post}} = \mathbf{H}^\top \mathbf{R}^{-1}z + \lambda^2 \mathbf{Q}_s^{-1}\boldsymbol{\mu}.$$

It is worth mentioning that the resulting posterior distribution is also Gaussian, with mean $s_{\text{post}}$ and covariance $\mathbf{Q}_{\text{post}}$, denoted as $s|z \sim \mathcal{N}(s_{\text{post}}, \mathbf{Q}_{\text{post}})$.

The reconstruction quality of (1) depends crucially on choosing appropriate covariance matrix parameters, or hyperparameters, that govern this prior (2) and the noise distribution of $\boldsymbol{\epsilon}$. In Section 3.1 we describe genHyBR methods for AIM where $\boldsymbol{\mu}$ is fixed but $\lambda$ is not known in advance.

In many applications, the prior mean $\boldsymbol{\mu}$ may also not be known in advance and must be estimated as a part of the inversion process. Some inverse models (commonly referred to as geostatistical inverse models) directly assimilate environmental data or data on emitting activities directly into the inverse model, and the relationships between the surface fluxes and these data are rarely known a priori. In other cases, an emissions inventory or bottom-up flux model may be biased too high or too low. In these cases, (2) no longer holds, violating the statistical assumptions of the inverse model. One workaround is to scale the inventory or flux model as part of the inverse modeling process, which we now describe.

**Unknown mean.** In cases where the prior mean is unknown, we can represent the prior information in the form of the hierarchical model

$$s|\boldsymbol{\beta} \sim \mathcal{N}(\mathbf{X}\boldsymbol{\beta}, \lambda^{-2}\mathbf{Q}_s), \qquad \boldsymbol{\beta} \sim \mathcal{N}(\boldsymbol{\mu}_\beta, \lambda_\beta^{-2}\mathbf{Q}_\beta). \tag{5}$$





Here $\mathbf{X} \in \mathbb{R}^{n \times p}$ is a fixed matrix that includes covariates (e.g., environmental data or activity data) or a bottom-up inventory/flux model, $\mathbf{Q}_s \in \mathbb{R}^{n \times n}$ is the prior covariance matrix, and $\lambda$ is a scaling parameter. A set of unknown coefficients $\boldsymbol{\beta} \in \mathbb{R}^p$ scale the columns of $\mathbf{X}$ and are estimated as part of the inverse model. These coefficients are assumed to follow a Gaussian distribution with given mean $\boldsymbol{\mu}_\beta \in \mathbb{R}^p$, covariance matrix $\mathbf{Q}_\beta \in \mathbb{R}^{p \times p}$, and scaling parameter $\lambda_\beta$.

Given the assumptions in (1) and (5), from Bayes' theorem the posterior probability density function for the unknown mean case can be written as

$$\pi(\boldsymbol{s}, \boldsymbol{\beta} | \boldsymbol{z}) \propto \pi(\boldsymbol{z}|\boldsymbol{s},\boldsymbol{\beta})\pi(\boldsymbol{s}|\boldsymbol{\beta})\pi(\boldsymbol{\beta})$$
$$\propto \exp\left( -\frac{1}{2}\|\mathbf{H}\boldsymbol{s} - \boldsymbol{z}\|^2_{\mathbf{R}^{-1}} - \frac{\lambda^2}{2}\|\boldsymbol{s} - \mathbf{X}\boldsymbol{\beta}\|^2_{\mathbf{Q}_s^{-1}} - \frac{\lambda_\beta^2}{2}\|\boldsymbol{\beta} - \boldsymbol{\mu}_\beta\|^2_{\mathbf{Q}_\beta^{-1}} \right). \tag{6}$$

The MAP estimate can be written as the solution of the optimization problem

$$\boldsymbol{\gamma}_{\text{post}} = \underset{\boldsymbol{\gamma} = [\boldsymbol{s}^\top, \boldsymbol{\beta}^\top]^\top}{\arg\min} \frac{1}{2}\|\mathbf{H}\boldsymbol{s} - \boldsymbol{z}\|^2_{\mathbf{R}^{-1}} + \frac{\lambda^2}{2}\|\boldsymbol{s} - \mathbf{X}\boldsymbol{\beta}\|^2_{\mathbf{Q}_s^{-1}} + \frac{\lambda_\beta^2}{2}\|\boldsymbol{\beta} - \boldsymbol{\mu}_\beta\|^2_{\mathbf{Q}_\beta^{-1}}. \tag{7}$$

The posterior distribution in (6) is Gaussian; therefore, the mean of the posterior distribution is also the MAP estimate and the covariance is the inverse of the Hessian matrix of (7) and is given by

$$\boldsymbol{\Gamma}_{\text{post}} = \begin{bmatrix} \lambda^2 \mathbf{Q}_s^{-1} + \mathbf{H}^\top \mathbf{R}^{-1} \mathbf{H} & -\lambda^2 \mathbf{Q}_s^{-1} \mathbf{X} \\ -\lambda^2 \mathbf{X}^\top \mathbf{Q}_s^{-1} & \lambda_\beta^2 \mathbf{Q}_\beta^{-1} + \lambda^2 \mathbf{X}^\top \mathbf{Q}_s^{-1} \mathbf{X} \end{bmatrix}^{-1}. \tag{8}$$

Therefore, the resulting posterior distribution is

$$\boldsymbol{\gamma} \mid \boldsymbol{z} \sim \mathcal{N}\left( \boldsymbol{\gamma}_{\text{post}}, \boldsymbol{\Gamma}_{\text{post}} \right). \tag{}$$

Here, $\lambda$ and $\lambda_\beta$ are scaling parameters that may not be known in advance, but we assume that $\lambda_\beta = \alpha\lambda$ with a constant $\alpha > 0$, where $\alpha$ is set in advance. We describe genHyBR methods for the unknown mean case in Section 3.3.

Note that previous works (Miller et al., 2020; Saibaba and Kitanidis, 2015) assume an improper prior for $\boldsymbol{\beta}$ (i.e., $p(\boldsymbol{\beta}) \propto 1$), in which case, a solution estimate can be obtained as $\widehat{\boldsymbol{s}} = \mathbf{X}\widehat{\boldsymbol{\beta}} + \mathbf{Q}_s \mathbf{H}^\top \widehat{\boldsymbol{\xi}}$, where

$$\begin{bmatrix} \mathbf{H}\mathbf{Q}_s \mathbf{H}^\top + \mathbf{R} & \mathbf{H}\mathbf{X} \\ (\mathbf{H}\mathbf{X})^\top & \mathbf{0} \end{bmatrix} \begin{bmatrix} \widehat{\boldsymbol{\xi}} \\ \widehat{\boldsymbol{\beta}} \end{bmatrix} = \begin{bmatrix} \boldsymbol{z} \\ \mathbf{0} \end{bmatrix}. \tag{9}$$

The system in (9) is often referred to as the dual function form, and there are several equivalent formulations of these equations (Michalak et al., 2004). The size of the resulting system of equations is $(m + p) \times (m + p)$, where $m$ is the number of measurements and $p$ is the number of unknown parameters in $\boldsymbol{\beta}$, so forming or inverting the matrix in (9) is infeasible in many applications. The approach taken in (Saibaba and Kitanidis, 2012; Miller et al., 2020) uses a matrix-free iterative method to solve (9); however, the number of required iterations can be very large, especially for problems with many measurements, and even with the use of a preconditioner.

In this paper, we follow a different approach to handle the unknown mean case by using iterative hybrid approaches on a reformulated problem. Since these methods work directly on the least-squares problem (7), the number of unknown parameters





is $n + p$. However, the size of the linear system that defines the MAP estimate is independent of the number of observations,
making it attractive for large datasets. Furthermore, our framework can handle a wide class of prior covariance operators, where
the resulting prior covariance matrices are large and dense and explicitly forming and factorizing these matrices is prohibitively
expensive. These include, for example, prior covariance matrices that arise from nonseparable, spatiotemporal covariance
kernels and parameterized kernels on non-uniform grids. Our approach only relies on forming matrix-vector products with the
covariance matrices, and is compatible with acceleration techniques using Fast Fourier Transform (FFT) or hierarchical matrix
approaches. See (Chung and Saibaba, 2017; Chung et al., 2018) for a detailed discussion. Thus, as we show in Section 4, our
approach can incorporate various prior models and can scale to very large data sets.

## 3 Generalized hybrid projection methods for AIM

In this section, we describe generalized hybrid projection methods, dubbed genHyBR methods, for AIM. Hybrid projection
methods were first developed in the 1980's as a way to combine iterative projection methods (e.g., Krylov subspace meth-
ods) and variational regularization methods (e.g., Tikhonov regularization) for solving very large inverse problems. These are
iterative methods, where each iteration requires the expansion of the solution subspace, the estimation of the regularization
parameter(s), and the solution of a projected, regularized problem. We point the interested reader to survey papers (Chung and
Gazzola, 2021; Gazzola and Sabaté Landman, 2020). In (Chung and Saibaba, 2017), genHyBR methods were developed for
computing Tikhonov regularized solutions to problems where explicit computations of the square root and inverse of the prior
covariance matrix are not feasible. This work enabled hybrid projection methods for more general regularization terms. The
main benefits of genHyBR methods that make them ideal for large large-scale AIM are *efficiency*, due to fast convergence to an
accurate reconstruction of surface fluxes where efficient matrix-vector multiplications are exploited at each iteration, *automatic*
estimation of parameters (e.g., hyperparameters and algorithmic parameters), and *flexibility* because they can be paired with
many different atmospheric models.

We describe genHyBR methods for both the fixed mean case (Section 3.1) and the unknown mean case (Section 3.3), with
particular emphasis on the associated challenges for large datasets and subsequent UQ (Section 3.2). We provide a general
overview of our approach, including the main components of genHyBR methods, in the flowchart in Figure 1.

### 3.1 Generalized hybrid methods with fixed mean

To introduce genHyBR methods, we begin with the fixed mean case described in Section 2. If symmetric decompositions
$\mathbf{R}^{-1} = \mathbf{L}_{\mathbf{R}}^{\top} \mathbf{L}_{\mathbf{R}}$ and $\mathbf{Q}_s^{-1} = \mathbf{L}_s^{\top} \mathbf{L}_s$ are available, then optimization problem (4) can be rewritten in the standard least-squares
form

$$\boldsymbol{s}_{\text{post}} = \underset{\boldsymbol{s} \in \mathbb{R}^n}{\arg\min} \frac{1}{2} \left\| \begin{bmatrix} \mathbf{L}_{\mathbf{R}} \mathbf{H} \\ \lambda \mathbf{L}_s \end{bmatrix} \boldsymbol{s} - \begin{bmatrix} \mathbf{L}_{\mathbf{R}} \boldsymbol{z} \\ \lambda \mathbf{L}_s \boldsymbol{\mu} \end{bmatrix} \right\|_2^2 .$$





**Figure 1.** This flowchart provides a general overview of using genHyBR methods for AIM and subsequent UQ. Given input (corresponding to the observations and details of the problem setup), genHyBR is an iterative approach to approximate the MAP estimate. Each iteration of genHyBR consists of expanding the solution subspace, projecting the problem, estimating a regularization parameter, and solving a projected, regularized problem. After obtaining the MAP estimate, information computed from genHyBR can be used to efficiently estimate the posterior variance for UQ.

However, computing $\mathbf{L}_s$ can be computationally infeasible for large $n$ and $\lambda$ may not be known a priori. This motivates us to use the following change of variables,

$$\boldsymbol{x} \leftarrow \mathbf{Q}_s^{-1}(\boldsymbol{s} - \boldsymbol{\mu}), \quad \boldsymbol{b} \leftarrow \boldsymbol{z} - \mathbf{H}\boldsymbol{\mu}, \tag{10}$$





in which case, the solution to problem (4) is given by $s_{\text{post}} = \boldsymbol{\mu} + \mathbf{Q}_s \boldsymbol{x}$ where $\boldsymbol{x}$ solves

$$\min_{\boldsymbol{x} \in \mathbb{R}^n} \frac{1}{2} \|\mathbf{H}\mathbf{Q}_s\boldsymbol{x} - \boldsymbol{b}\|^2_{\mathbf{R}^{-1}} + \frac{\lambda^2}{2} \|\boldsymbol{x}\|^2_{\mathbf{Q}_s}. \tag{11}$$

Note that, with this reformulation, we avoid $\mathbf{L}_s$, $\mathbf{L}_s^{-1}$, and $\mathbf{Q}_s^{-1}$ and only require matrix-vector products with $\mathbf{Q}_s$. Furthermore, for iterative methods for (11), the matrix $\mathbf{H}$ does not need to be formed explicitly, as we only need access to matrix-vector and

matrix-transpose-vector products.

There are two main ingredients in the genHyBR approach: (1) the generalized Golub-Kahan bidiagonalization approach for constructing a solution subspace and (2) regularization parameter estimation methods for computing a suitable regularization parameter in the projected space.

### 3.1.1 Generalized Golub-Kahan bidiagonalization

We now describe the generalized Golub-Kahan bidiagonalization approach that is the backbone of the genHyBR method. Given matrices $\mathbf{H}$, $\mathbf{R}$, $\mathbf{Q}_s$ and vector $\boldsymbol{b}$ from (11), the basic idea is to generate a set of basis vectors contained in $\mathbf{V}_k$ for the Krylov subspace,

$$\mathcal{S}_k \equiv \mathcal{R}(\mathbf{V}_k) = \mathcal{K}_k(\mathbf{H}^\top \mathbf{R}^{-1} \mathbf{H} \mathbf{Q}_s, \mathbf{H}^\top \mathbf{R}^{-1} \boldsymbol{b})$$

where $\mathcal{R}(\cdot)$ denotes the column space and the Krylov subspace is $\mathcal{K}_k(\mathbf{M}, \boldsymbol{f}) = \text{Span}\{\boldsymbol{f}, \mathbf{M}\boldsymbol{f}, \ldots, \mathbf{M}^{k-1}\boldsymbol{f}\}$. The generated basis

vectors span a low-dimensional subspace that is rich in information about important directions; thus, solutions to the (smaller) projected problem often provide good approximations to the solution of the high-dimensional problem. With initializations $\delta_1 = \|\boldsymbol{b}\|_{\mathbf{R}^{-1}}, \boldsymbol{u}_1 = \boldsymbol{b}/\delta_1$ and $\gamma_1 \boldsymbol{v}_1 = \mathbf{H}^\top \mathbf{R}^{-1} \boldsymbol{u}_1$, the $k$th iteration of the generalized Golub-Kahan bidiagonalization procedure generates vectors $\boldsymbol{u}_{k+1}$ and $\boldsymbol{v}_{k+1}$ such that

$$\gamma_{k+1}\boldsymbol{u}_{k+1} = \mathbf{H}\mathbf{Q}_s\boldsymbol{v}_k - \gamma_k\boldsymbol{u}_k$$

$$\delta_{k+1}\boldsymbol{v}_{k+1} = \mathbf{H}^\top \mathbf{R}^{-1}\boldsymbol{u}_{k+1} - \delta_{k+1}\boldsymbol{v}_k$$

where scalars $\gamma_k, \delta_k \geq 0$ are computed such that $\|\boldsymbol{u}_k\|_{\mathbf{R}^{-1}} = \|\boldsymbol{v}_k\|_{\mathbf{Q}_s} = 1$. At the end of $k$ iterations, we have

$$\mathbf{B}_k \equiv \begin{bmatrix} \gamma_1 & 0 & \cdots & 0 \\ \delta_2 & \gamma_2 & \ddots & \vdots \\ 0 & \delta_3 & \ddots & 0 \\ \vdots & \ddots & \ddots & \gamma_k \\ 0 & \cdots & 0 & \delta_{k+1} \end{bmatrix}, \quad \mathbf{U}_{k+1} \equiv \begin{bmatrix} \boldsymbol{u}_1, \ldots, \boldsymbol{u}_{k+1} \end{bmatrix}, \quad \text{and} \quad \mathbf{V}_k \equiv \begin{bmatrix} \boldsymbol{v}_1, \ldots, \boldsymbol{v}_k \end{bmatrix},$$

where the following relations hold

$$\mathbf{H}\mathbf{Q}_s\mathbf{V}_k = \mathbf{U}_{k+1}\mathbf{B}_k \quad \text{and} \quad \mathbf{H}^\top \mathbf{R}^{-1}\mathbf{U}_{k+1} = \mathbf{V}_k\mathbf{B}_k^\top + \gamma_{k+1}\boldsymbol{v}_{k+1}\boldsymbol{e}_{k+1}^\top, \tag{12}$$



where $\boldsymbol{e}_j$ is the $j$th column of an identity matrix with the appropriate dimensions. Also, matrices $\mathbf{U}_{k+1}$ and $\mathbf{V}_k$ satisfy the following orthogonality conditions

$$\mathbf{U}_{k+1}^\top \mathbf{R}^{-1} \mathbf{U}_{k+1} = \mathbf{I}_{k+1} \quad \text{and} \quad \mathbf{V}_k^\top \mathbf{Q}_s \mathbf{V}_k = \mathbf{I}_k, \tag{13}$$

with $\mathbf{U}_{k+1}\delta_1 \boldsymbol{e}_1 = \boldsymbol{b}$. Then, for given $\lambda > 0$ the solution to (11) is recovered by $\boldsymbol{x}_{k,\lambda} = \mathbf{Q}_s \mathbf{V}_k \boldsymbol{y}_{k,\lambda}$ where $\boldsymbol{y}_{k,\lambda}$ is the solution to the regularized, projected problem

$$\min_{\boldsymbol{y} \in \mathbb{R}^k} \frac{1}{2}\|\mathbf{B}_k\boldsymbol{y} - \delta_1 \boldsymbol{e}_1\|_2^2 + \frac{\lambda^2}{2}\|\boldsymbol{y}\|_2^2. \tag{14}$$

Notice that (14) is a standard least-squares problem with Tikhonov regularization, and since the coefficient matrix $\mathbf{B}_k$ is of size $(k+1) \times k$, the solution can be computed efficiently (Björck, 1996). Each iteration of the generalized Golub-Kahan bidiagonalization process requires one matrix-vector product with $\mathbf{H}$ and its adjoint (suppose we denote its cost by $T_\mathbf{H}$), two matrix-vector products with $\mathbf{Q}_s$ (similarly, denoted $T_{\mathbf{Q}_s}$) and additional $\mathcal{O}(m+n)$ floating point operations (flops). To compute

the solution of the least-squares problem (14), the cost is $\mathcal{O}(k^3)$ flops, and the cost of forming $\boldsymbol{x}_{k,\lambda}$ is $\mathcal{O}(nk)$ flops. Thus, the overall cost of the algorithm is

$$T_{\text{genGK}} = 2k(T_\mathbf{H} + T_{\mathbf{Q}_s}) + \mathcal{O}(k(m+n)) \text{ flops.}$$

In practice, the vectors $\{\boldsymbol{u}_k\}$ and $\{\boldsymbol{v}_k\}$ lose orthogonality in floating point arithmetic, so full or partial re-orthogonalization (Barlow, 2013) can be used to ensure orthogonality. This costs an additional $\mathcal{O}(k^2(m+n))$ flops. Thus far we have described

an iterative method for approximating the MAP estimate (4), where the $k$th iterate is given by $\boldsymbol{s}_{k,\lambda} = \boldsymbol{\mu} + \mathbf{Q}_s \boldsymbol{x}_{k,\lambda}$ for fixed regularization parameter $\lambda$.

### 3.1.2   Regularization parameter estimation methods

In this subsection, we highlight one of the main computational benefits of hybrid projection methods, which is the ability to estimate regularization parameters efficiently and adaptively, while still ensuring robustness of the solution. For genHyBR

approaches, we use the generalized Golub-Kahan bidiagonalization to generate a projection subspace, and solve the projected problem (14), while simultaneously estimating the regularization parameter $\lambda$. Notice that the regularization term in the projected system (14) is standard Tikhonov regularization, and a plethora of parameter estimation methods exist for Tikhonov regularization (Bardsley, 2018; Hansen, 2010). Here we focus on the discrepancy principle (DP) and point the interested reader to Appendix A for further details on other parameter estimation methods can be incorporated within genHyBR methods

for AIM.

The discrepancy principle (DP) is a common approach for estimating a regularization parameter, where the main goal is to determine $\lambda$ such that the residual norm for the regularized reconstruction matches a given estimate of the noise level in the observations. That is, the DP method selects the largest parameter value $\lambda$ for which the reconstructed fluxes $\boldsymbol{s}_\lambda$ satisfy

$$\mathcal{D}_{\text{full}}(\lambda) = \|\mathbf{H}\boldsymbol{s}_\lambda - \boldsymbol{z}\|_{\mathbf{R}^{-1}}^2 \leq \tau m, \tag{15}$$





where $\tau \geq 1$ is a user-defined parameter and $m$ is the expected value of $\|\boldsymbol{\epsilon}\|^2_{\mathbf{R}^{-1}}$. Typical choices for safety factor $\tau$ are in the range $1 \leq \tau \leq 2$.

For a given $\lambda$, evaluating $\mathcal{D}_{\mathrm{full}}(\lambda)$ requires computing $\boldsymbol{s}_\lambda$ and matrix-vector multiplication with $\mathbf{H}$, which can get costly if many different values of $\lambda$ are desired. However, using the relationships in (12), the residual norm can be simplified as

$$\|\mathbf{H}\boldsymbol{s}_{k,\lambda} - \boldsymbol{z}\|^2_{\mathbf{R}^{-1}} = \|\mathbf{B}_k\boldsymbol{y}_{k,\lambda} - \delta_1\boldsymbol{e}_1\|^2_2 \equiv \mathcal{D}_{\mathrm{proj}}(\lambda). \tag{16}$$

Thus, we let $\lambda_k$ be the regularization parameter estimated for the projected problem at the $k$th iteration, such that $\mathcal{D}_{\mathrm{proj}}(\lambda_k) \leq \tau m$. Then as the number of iterations $k$ increases, the estimated DP regularization parameter for the projected problem becomes a better approximation of the DP parameter for the original problem.

The advantage of this approach is two-fold: the regularization parameter is selected adaptively (i.e., each iteration can have a different regularization parameter), and the cost of parameter selection is cheap ($\mathcal{O}(k^3)$ flops) since we work with small matrices of size $(k+1)\times k$ and $k$ is much smaller than $m$ and $n$. Furthermore, there are various theoretical results that show that selecting the regularization parameter for the projected problem (i.e., project-then-regularize) is equivalent to first estimating the regularization parameter and then using an iterative projection method (i.e., regularize-then-project) (Chung and Gazzola, 2021).

## 3.2 Approximation to the posterior covariance matrices

In the Bayesian approach for fixed parameter $\lambda$, the posterior distribution (3) is Gaussian and, thus, is fully specified by the mean and covariance matrix. However, neither computing nor storing the covariance matrix is feasible, making further uncertainty estimation challenging. Instead, we follow the approach described in (Chung et al., 2018; Saibaba et al., 2020) for the fixed mean case, where an approximation to the posterior covariance matrix is obtained using the computed vectors generated using the generalized Golub-Kahan bidiagonalization process. An advantage of this approach is that, by storing partial information while computing the MAP estimate, we can approximately compute the uncertainty associated with the MAP estimate (e.g., posterior variance) with minimal additional cost, and no further accesses to the forward and adjoint models. For the fixed mean case, we refer to this approach as `genHyBRs` with UQ and provide a summary in Algorithm 1.

In the following, we provide some details regarding the estimation of the posterior variance in the known mean case. This material has previously appeared in (Chung et al., 2018, Section 4.1), but we provide a brief description here for completeness. An alternative expression for the posterior covariance is

$$\mathbf{Q}_{\mathrm{post}} = \mathbf{Q}_s(\lambda^2\mathbf{Q}_s + \mathbf{Q}_s\mathbf{H}^\top\mathbf{R}^{-1}\mathbf{H}\mathbf{Q}_s)^{-1}\mathbf{Q}_s,$$

which is obtained by factoring out $\mathbf{Q}_s$. This expression is not computationally feasible for large inverse problems but can be approximated using the outputs of the generalized Golub-Kahan bidiagonalization (described in Subsection 3.1.1). After $k$ steps of the generalized Golub-Kahan bidiagonalization approach, we have matrices $\mathbf{U}_{k+1}, \mathbf{V}_k$, and $\mathbf{B}_k$. Let $\mathbf{B}_k^\top\mathbf{B}_k = \mathbf{W}_k\mathbf{\Theta}_k\mathbf{W}_k^\top$ be the eigenvalue decomposition with eigenvalues $\theta_1,\ldots,\theta_k$. Next, we compute the matrix $\mathbf{Z}_k = \mathbf{Q}_s\mathbf{V}_k\mathbf{W}_k$ and the diagonal





---

**Algorithm 1** AIM with fixed mean—`genHyBRs` with UQ

---

**Require:** Matrices $\mathbf{H}$, $\mathbf{R}$ and $\mathbf{Q}$, and vector $\boldsymbol{b}$.

1: {/∗ Compute MAP estimate ∗/}

2: initialize $\boldsymbol{u}_1 = \boldsymbol{b}/\|\boldsymbol{b}\|_{\mathbf{R}^{-1}}$

3: **for** $j = 1, \ldots, k$ **do**

4:     one iteration of generalized Golub-Kahan bidiagonalization to obtain $\mathbf{B}_j$, $\mathbf{U}_{j+1}$, and $\mathbf{V}_j$

5:     estimate regularization parameter $\lambda$ and compute $\boldsymbol{x}_{j,\lambda} = \mathbf{Q}_s \mathbf{V}_j \boldsymbol{y}_{j,\lambda}$ where $\boldsymbol{y}_{j,\lambda}$ solves (14)

6: **end for**

7: compute the MAP estimate $\boldsymbol{s}_{k,\lambda} = \boldsymbol{\mu} + \mathbf{Q}_s \boldsymbol{x}_{k,\lambda}$

8: {/∗ Compute the approximation to the posterior variance ∗/}

9: compute the eigendecomposition $\mathbf{B}_k^\top \mathbf{B}_k = \mathbf{W}_k \boldsymbol{\Theta}_k \mathbf{W}_k^\top$

10: compute $\mathbf{Z}_k = \mathbf{Q}_s \mathbf{V}_k \mathbf{W}_k$ and $\boldsymbol{\Delta}_k = \lambda^{-2} \mathrm{diag}(\frac{\theta_1}{\theta_1 + \lambda^2}, \ldots, \frac{\theta_k}{\theta_k + \lambda^2})$

11: compute $\boldsymbol{d}_{\mathrm{LR}} = \mathrm{LowRankDiag}(\mathbf{Z}_k \boldsymbol{\Delta}_k, \mathbf{Z}_k)$ using Algorithm 2 and $\boldsymbol{d}_s = \mathrm{diag}(\mathbf{Q}_s)$

12: estimate diagonal of approximate posterior covariance matrix $\boldsymbol{d}_{k,\lambda} = \lambda^{-2} \boldsymbol{d}_s - \boldsymbol{d}_{\mathrm{LR}}$

13: **return** MAP estimate $\boldsymbol{s}_{k,\lambda}$ and variance estimate $\boldsymbol{d}_{k,\lambda}$

---

matrix

$$\boldsymbol{\Delta}_k = \lambda^{-2} \begin{bmatrix} \frac{\theta_1}{\theta_1 + \lambda^2} & & \\ & \ddots & \\ & & \frac{\theta_k}{\theta_k + \lambda^2} \end{bmatrix} \in \mathbb{R}^{k \times k}.$$

Then we can approximate the posterior covariance matrix as

$$\widetilde{\mathbf{Q}}_{\mathrm{post}} = \mathbf{Q}_s (\lambda^2 \mathbf{Q}_s + \mathbf{Z}_k \boldsymbol{\Theta}_k \mathbf{Z}_k^\top)^{-1} \mathbf{Q}_s = \lambda^{-2} \mathbf{Q}_s - \mathbf{Z}_k \boldsymbol{\Delta}_k \mathbf{Z}_k^\top,$$

where the error in the approximation was analyzed in (Saibaba et al., 2020).

To visualize the uncertainty, it is common to compute the diagonals of the posterior covariance (known as the posterior variance). The diagonals of $\mathbf{Q}_s$ are typically known analytically; the diagonals of the low-rank term $\mathbf{Z}_k \boldsymbol{\Delta}_k \mathbf{Z}_k^\top$ can be computed efficiently using Algorithm 2 with input $\mathbf{Y} = \mathbf{Z}_k \boldsymbol{\Delta}_k$ and $\mathbf{Z} = \mathbf{Z}_k$. An important point worth emphasizing is that the approximation to the posterior covariance need not be computed explicitly. More precisely, in addition to storing the information required for storing $\mathbf{Q}_s$, we only need to store $nk + k$ additional entries corresponding to the matrices $\mathbf{Z}_k$ and $\boldsymbol{\Delta}_k$.

In addition, one can also use `genHyBRs` to compute the posterior variance of the sum of the fluxes (or analogously, the variance of the mean). To do so, let $\mathbf{1}$ denote an $n \times 1$ vector of ones and multiply the components of $\widetilde{\mathbf{Q}}_{\mathrm{post}}$ as

$$\mathbf{1}^\top \widetilde{\mathbf{Q}}_{\mathrm{post}} \mathbf{1} = \lambda^{-2} (\mathbf{1}^\top \mathbf{Q}_s \mathbf{1}) - (\mathbf{1}^\top \mathbf{Z}_k) \boldsymbol{\Delta}_k (\mathbf{1}^\top \mathbf{Z}_k)^\top.$$

Several existing studies (Yadav and Michalak, 2013; Miller et al., 2020) describe how to efficiently compute $\mathbf{1}^\top \mathbf{Q}_s \mathbf{1}$ using

Kronecker products.





Note that in previous works (Chung et al., 2018; Saibaba et al., 2020), we found that additional reorthogonalization of the generalized Golub-Kahan basis vectors yielded more accurate results, so we perform them in the numerical experiments.

---

**Algorithm 2** Compute the diagonals of the low-rank term $\mathbf{Y}\mathbf{Z}^\top$. Call as $[\boldsymbol{d}] =$LowRankDiag$(\mathbf{Y}, \mathbf{Z})$

---

**Require:** Matrices $\mathbf{Y}, \mathbf{Z} \in \mathbb{R}^{n \times k}$ defining the outer product $\mathbf{Y}\mathbf{Z}^\top$

1: **for** $i = 1, \dots, n$ **do**

2:    $d_i = \sum_{j=1}^k \mathbf{Y}_{ij} \mathbf{Z}_{ij}$

3: **end for**

4: **return** vector $\boldsymbol{d} \in \mathbb{R}^n$ containing the diagonals of $\mathbf{Y}\mathbf{Z}^\top$

---

### 3.3 Hierarchical Gaussian priors: Reformulation for mean estimation

Next we describe genHyBR methods for AIM with unknown mean as described in Section 2 with assumptions given in (5). First, we reformulate the problem for simultaneous estimation of the surface fluxes in $\boldsymbol{s}$ and the covariate parameters in $\boldsymbol{\beta}$. Then, we describe how to use genHyBR methods for computing the corresponding MAP estimate (7) and for subsequent UQ. We refer to this approach as `genHyBRmean` with UQ, and since the derivations follow those in Sections 3.1 and 3.2, specific details have been relegated to Appendix B. However, we would like to emphasize that this derivation is new and a contribution of this work.

Notice that optimization problem (7) can be rewritten in standard least-squares form

$$\boldsymbol{\gamma}_{\text{post}} = \underset{\boldsymbol{\gamma} = [\boldsymbol{s}^\top, \boldsymbol{\beta}^\top]^\top}{\arg\min} \frac{1}{2} \left\| \begin{bmatrix} \mathbf{L_R H} & \mathbf{0} \\ \lambda \mathbf{L}_s & -\lambda \mathbf{L}_s \mathbf{X} \\ \mathbf{0} & \lambda_\beta \mathbf{L}_\beta \end{bmatrix} \begin{bmatrix} \boldsymbol{s} \\ \boldsymbol{\beta} \end{bmatrix} - \begin{bmatrix} \mathbf{L_R} z \\ \mathbf{0} \\ \lambda_\beta \mathbf{L}_\beta \boldsymbol{\mu}_\beta \end{bmatrix} \right\|_2^2 ,$$

if symmetric decompositions $\mathbf{R}^{-1} = \mathbf{L_R}^\top \mathbf{L_R}, \mathbf{Q}_s^{-1} = \mathbf{L}_s^\top \mathbf{L}_s$, and $\mathbf{Q}_\beta^{-1} = \mathbf{L}_\beta^\top \mathbf{L}_\beta$ are available. Since computing $\mathbf{L}_s$ can be computationally infeasible (e.g., for spatiotemporal covariance matrices where $n$ is large), we propose a similar change of variables to avoid $\mathbf{L}_s$,

$$\widetilde{s} \leftarrow \boldsymbol{s} - \mathbf{X}\boldsymbol{\mu}_\beta, \quad \widetilde{\boldsymbol{\beta}} \leftarrow \boldsymbol{\beta} - \boldsymbol{\mu}_\beta, \quad \widetilde{z} \leftarrow z - \mathbf{H}\mathbf{X}\boldsymbol{\mu}_\beta. \tag{17}$$

For notational convenience, we define the concatenated vector $\widetilde{\boldsymbol{\gamma}} = [\widetilde{s}^\top, \widetilde{\boldsymbol{\beta}}^\top]^\top$ and let $\mathbf{K} = \begin{bmatrix} \mathbf{H} & \mathbf{0} \end{bmatrix} \in \mathbb{R}^{m \times (n+p)}$. Then, optimization problem (7) can be written as

$$\min_{\widetilde{\boldsymbol{\gamma}} \in \mathbb{R}^{n+p}} \frac{1}{2} \| \mathbf{K}\widetilde{\boldsymbol{\gamma}} - \widetilde{z} \|_{\mathbf{R}^{-1}}^2 + \frac{\lambda^2}{2} \| \widetilde{\boldsymbol{\gamma}} \|_{\mathbf{Q}^{-1}}^2 . \tag{18}$$

where

$$\mathbf{Q} = \begin{bmatrix} \mathbf{Q}_s + \frac{1}{\alpha^2} \mathbf{X}\mathbf{Q}_\beta \mathbf{X}^\top & \frac{1}{\alpha^2} \mathbf{X}\mathbf{Q}_\beta \\ \frac{1}{\alpha^2} (\mathbf{X}\mathbf{Q}_\beta)^\top & \frac{1}{\alpha^2} \mathbf{Q}_\beta \end{bmatrix} = \begin{bmatrix} \mathbf{Q}_s & \mathbf{0} \\ \mathbf{0} & \mathbf{0} \end{bmatrix} + \frac{1}{\alpha^2} \begin{bmatrix} \mathbf{X} \\ \mathbf{I} \end{bmatrix} \mathbf{Q}_\beta \begin{bmatrix} \mathbf{X}^\top & \mathbf{I} \end{bmatrix} . \tag{19}$$





Derivations are provided in Appendix B1. Note that, in practice, neither of the matrices $\mathbf{H}$ nor $\mathbf{K}$ need to be formed explicitly since we only need access to matrix-vector products with these matrices and their transposes. Also we do not explicitly construct $\mathbf{Q}$, but instead provide an efficient way to form matrix-vector products with $\mathbf{Q}$.

In summary, to handle AIM with unknown mean, the genHyBR method can be used to solve (18) (as described in Section 3.1 with $\mathbf{Q}$ instead of $\mathbf{Q}_s$, $\mathbf{K}$ instead of $\mathbf{H}$, and $\widetilde{z}$ instead of $b$) to efficiently obtain the solution $\widetilde{\gamma}_{k,\lambda} = [\widetilde{s}_{k,\lambda}^\top, \widetilde{\beta}_{k,\lambda}^\top]^\top$. Then, we recover the MAP estimate for $s$ and $\beta$ as

$$\gamma_{\text{post}} \approx \gamma_{k,\lambda} := \begin{bmatrix} \widetilde{s}_{k,\lambda} + \mathbf{X}\mu_\beta \\ \widetilde{\beta}_{k,\lambda} + \mu_\beta \end{bmatrix}. \tag{20}$$

Similar to the fixed mean case, we can efficiently approximate the posterior covariance matrix and its diagonals using elements of the generalized Golub-Kahan bidiagonalization algorithm, as described in Appendix B3. We remark that efficient UQ approaches for the unknown mean case were considered in (Saibaba and Kitanidis, 2015), but our approach differs in that we reuse information contained in the subspaces generated during the iterative method, rather than randomization techniques, making these derivations straightforward and the approaches widely applicable.

## 4 Numerical results

We evaluate the inverse modeling algorithms described in this paper using the two case studies described in Section 4.1. For the numerical experiments, we denote the two methods we test as

- `genHyBRs` refers to the fixed mean case, and involves solving the optimization problem (4), using the approach described in Sections 3.1.1 and 3.1.2,

- `genHyBRmean` refers to the unknown mean case, and involves solving the optimization problem (18), using the approach described in Section 3.3.

For comparison, we use a direct inversion method, which solves (9) using MATLAB's "backslash" operator. Numerical experiments presented here were obtained using MATLAB on a compute server with four Intel 15 core 2.8 GHz processors and 1 TB of RAM.

### 4.1 Overview of the case studies

We explore two case studies on estimating $CO_2$ fluxes across North America using observations from NASA's OCO-2 satellite. In the first case study, we estimate $CO_2$ fluxes for 6 weeks (late June through July 2015), an inverse problem that is small enough to estimate using the direct method. The second case study using one year of observations (Sept. 2014 − Aug. 2015) is too large to estimate directly on many or most computer systems. The goal of these experiments is to demonstrate the performance of the generalized hybrid methods for solving the inverse problem with automatic parameter selection.

The case studies explored here parallel those in (Miller et al., 2020) and (Liu et al., 2020). We provide an overview of these case studies, but refer to (Miller et al., 2020) for additional detail on the specific setup. Both of the case studies use synthetic





OCO-2 observations that are generated using $CO_2$ fluxes from NOAA's CarbonTracker product (version 2019b). As a result of this setup, the true $CO_2$ fluxes ($s$) are known, making it easier to evaluate the accuracy of the algorithms tested here. All atmospheric transport simulations are from the Weather Research and Forecasting (WRF) Stochastic Time-Inverted Lagrangian Transport Model (STILT) modeling system (Lin et al., 2003; Nehrkorn et al., 2010). These simulations were generated as part

of NOAA's CarbonTracker-Lagrange program (Hu et al., 2019; Miller et al., 2020). Note that the WRF-STILT outputs can be used to explicitly construct $\mathbf{H}$, making it straightforward to calculate the direct solution to the inverse problem in the 6 week case study. By contrast, many inverse modeling studies use the adjoint of an Eulerian model. These modeling frameworks rarely produce an explicit $\mathbf{H}$ but instead output the product of $\mathbf{H}$ or $\mathbf{H}^\top$ and a vector. Though we use WRF-STILT for the case studies presented here, the genHyBR algorithms could also be paired with the the adjoint of an Eulerian model.

The goal is to estimate $CO_2$ fluxes at a 3-hourly temporal resolution and a $1° \times 1°$ latitude-longitude spatial resolution. Using this modeling framework, synthetic observations were obtained as in (1) by adding white Gaussian noise (representing measurement and model errors) to the output from the atmospheric transport model. In total, there are $m = 1.92 \times 10^4$ synthetic observations and $n = 1.06 \times 10^6$ unknown $CO_2$ fluxes in the 6 week case study. By contrast, for the much larger one year case study, there are $m = 9.9 \times 10^4$ synthetic observations and $n = 9.4 \times 10^6$ unknown $CO_2$ fluxes to be estimated.

The noise covariance matrix is structured as $\mathbf{R} = \sigma^2 \mathbf{I}$ where $\sigma^2$ represents the noise variance. In this case, the discrepancy principle formula simplifies to

$$\mathcal{D}_{\text{full}}(\lambda) = \|\mathbf{H}s_\lambda - z\|_2^2 \leq \tau m \sigma^2.$$

Notice that the DP approach requires *a priori* knowledge or an estimate of the noise variance $\sigma^2$. In (Miller et al., 2020), $\sigma = 2$ was used, which leads to a relatively large amount of noise in the observations. We test different values of $\sigma$, corresponding

to different noise levels (referred to as `nlevel`), as shown in Table 1. We note that some of the considered noise levels, although very high compared to examples in the inverse problems literature, are lower than typically observed in practice for atmospheric inverse problems. However, recent studies (Miller et al., 2018; O'Dell et al., 2018; Crowell et al., 2019; Miller and Michalak, 2020) show that errors in OCO-2 observations have been gradually decreasing with regular improvements in the satellite retrieval algorithms and bias corrections. Some of the values in Table 1 are low, even considering these recent

improvements. With that said, these values are aspirational and may become more realistic in the future as observational and atmospheric modeling errors decline. Furthermore, they provide an opportunity to explore the behavior of the proposed inverse modeling algorithms at many different error levels.

Next, we describe the prior used in both case studies. The prior flux estimate is set to a constant value for the case studies explored here. As a result of this setup, the prior flux estimate does not contain any spatiotemporal patterns, and the patterns

in the posterior fluxes solely reflect the information content of the atmospheric observations. For the cases with a fixed mean (`genHyBRs`), we set $\boldsymbol{\mu} = \mathbf{0}$, as has been done in several existing studies on inverse modeling algorithms (Rodgers, 2000; Chung and Saibaba, 2017). For the unknown mean case (`genHyBRmean`), the setup for $\mathbf{X}$ is identical to that used in (Miller et al., 2020). Specifically, for the 6 week case study, $\mathbf{X}$ has eight columns, and each column corresponds to a different 3-hourly time period of the day. A given column of $\mathbf{X}$ contains values of one for all flux elements that correspond to a given 3-hourly





time of day and zero for all other elements. $CO_2$ fluxes have a large diurnal cycle, and this setup accounts for the fact that $CO_2$ fluxes at different times of day will have a different mean. In the 1 year case study, $\mathbf{X}$ has 12 columns, and each column corresponds to fluxes in a different month of the year (Miller et al., 2020).

For the prior covariance matrix of unknown fluxes, $\mathbf{Q}_s = \mathbf{Q}_t \otimes \mathbf{Q}_g$ where $\mathbf{Q}_t$ represents the temporal covariance and $\mathbf{Q}_g$ represents the spatial covariance in the fluxes. The symbol $\otimes$ denotes the Kronecker product. We use a spherical covariance model for the spatial and temporal covariance. A spherical model is ideal because it decays to zero at the correlation length or time, and the resulting matrices are usually sparse:

$$k_t(d_t;\theta_t) = \begin{cases} 1 - \frac{3}{2}\left(\frac{d_t}{\theta_t}\right) + \frac{1}{2}\left(\frac{d_t}{\theta_t}\right)^3 & \text{if } d_t \le \theta_t, \\ 0 & \text{if } d_t > \theta_t, \end{cases} \tag{21}$$

$$k_g(d_g;\theta_g) = \begin{cases} 1 - \frac{3}{2}\left(\frac{d_g}{\theta_g}\right) + \frac{1}{2}\left(\frac{d_g}{\theta_g}\right)^3 & \text{if } d_g \le \theta_g, \\ 0 & \text{if } d_g > \theta_g, \end{cases} \tag{22}$$

where $d_t$ is the temporal difference, $d_g$ is the spherical distance, and $\theta_t, \theta_g$ represent the decorrelation time and decorrelation length, respectively. For the 6 week case study, we set $\theta_t = 9.854$ and $\theta_g = 555.42$, as in (Miller et al., 2020). The sparsity patterns of these covariance matrices are provided in Figure 2. For the one year study, we use slightly different parameters, as listed in full detail in (Miller et al., 2020). Notably, the variance is different in each month to better capture the impact of seasonal changes on the variability of $CO_2$ fluxes. Note that for the fixed mean case, the covariance matrices are sparse can be efficiently represented in factored form. However, the hybrid approaches proposed here can handle much more complicated cases (see e.g., (Chung and Saibaba, 2017; Chung et al., 2018)).

For the unknown mean case, the covariance matrix $\mathbf{Q}_\beta$ is set to be the identity matrix, and $\alpha = 10$ is used in the numerical experiments. We experimented with various choices for $\alpha$ and observed consistently good results with $\alpha = 10$. In all of the numerical experiments, we use the DP approach to select the regularization parameter within genHyBR methods with $\tau = 1$.

In subsequent discussion of the case study results, we provide relative reconstruction error norms computed as $\|\boldsymbol{s}_{k,\lambda} - \boldsymbol{s}\|_2 / \|\boldsymbol{s}\|_2$, where $\boldsymbol{s}$ denotes the true fluxes and $\boldsymbol{s}_{k,\lambda}$ contains the reconstructed spatiotemporal fluxes at the $k$th iteration.

## 4.2 Results of the case studies

Both the `genHyBRs` and `genHyBRmean` methods converge quickly and yield accurate estimates of the $CO_2$ fluxes relative to other inverse modeling methods. For the 6 weeks case study, Figure 3 shows the relative reconstruction error norms for

| nlevel | $\sigma\,(\mu\,\mathrm{mol}\,m^{-2}s^{-1})$ |
|--------|-----------|
| 5% | 0.0565 |
| 10% | 0.1134 |
| 50% | 0.5648 |

**Table 1.** Noise level and corresponding noise standard deviation $\sigma$ used in the 6-weeks case study experiments.





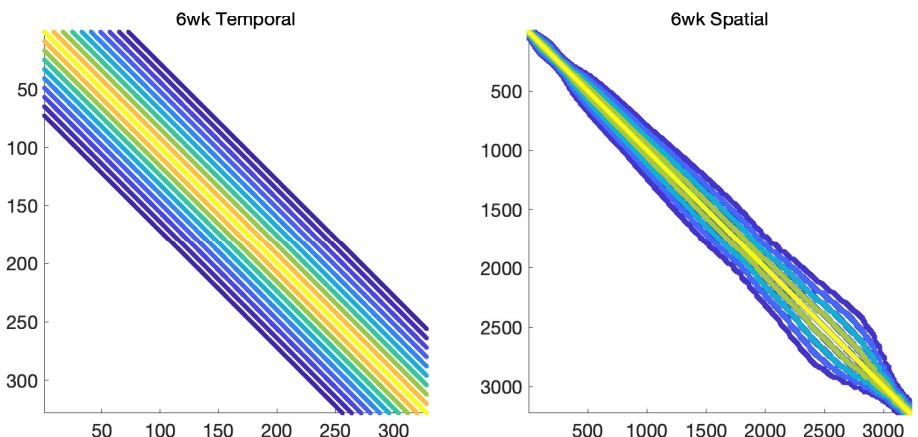

**Figure 2.** Sparsity pattern of the prior covariance matrices $\mathbf{Q}_t$ and $\mathbf{Q}_g$ for the 6 weeks case study.

both `genHyBRs` and `genHyBRmean` for three different noise levels and for different options for selecting the regularization
parameter. DP corresponds to the discrepancy principle and is an automatic approach that depends on the data and noise
level. The optimal regularization parameter, which corresponds to selecting the regularization parameter at each iteration that
minimizes the reconstruction error, is provided for comparison, although it is not obtainable in practice. All of the plots
with "none" correspond to $\lambda = 0$ and show *semiconvergent behavior*, which is revealed in the "U"-shape of the relative error
plot. That is, the relative reconstruction error norms decrease in early iterations, but increase in later iterations due to noise
contamination in the reconstructions. We observe that for all noise levels, genHyBR methods with regularization parameter
estimation (`genHyBRs-opt` and `genHyBRs-dp`) result in reconstruction error norms that decrease and flatten, thereby
overcoming the semiconvergent behavior of genHyBR methods with no regularization (`genHyBRs-none`). For reference, we
mark with horizontal lines the relative reconstruction error norm for direct reconstructions for two values of $\lambda$. Recall that the
direct method requires $\lambda$ to be fixed in advance, and reconstructions for $\lambda\sigma = 1$ do not yield accurate reconstructions of the
$CO_2$ fluxes. Using the optimal regularization parameter computed from `genHyBRmean` (these values are provided in Table
2), we show that a good reconstruction can be obtained with the direct method if a good regularization parameter is available,
but obtaining this result may require careful and expensive tuning.

We also find that the algorithm that simultaneously estimates the mean (`genHyBRmean`) yields lower errors than the al-
gorithm with a fixed mean (`genHyBRs`). Notably, the difference in performance between these two algorithms grows as the
noise level increases. In other words, the comparative advantage of `genHyBRmean` is even larger at higher noise levels. This
result implies that mean estimation becomes critical for problems with large noise levels.

The estimated fluxes also exhibit spatial patterns that mirror fluxes estimated using a direct (e.g., analytical) approach. Maps
of $CO_2$ fluxes, averaged over 6 weeks and corresponding to 50% noise level, are shown in Figure 4. We provide the true average
flux, the `genHyBRs` and `genHyBRmean` reconstructions for various parameter choices, and the direct method reconstruction





using the optimal regularization parameter from `genHyBRmean`. Notice that these relative reconstruction error values are different than those provided in Figure 3 because they represent error norms computed on the average image rather than on the native 3-hour resolution of the estimated fluxes. Also, to better highlight broad spatial patterns, the colormap has been constrained so that average flux estimates above 2 are set to 2 and estimates below $-5$ are set to $-5$.

Perhaps surprisingly, the genHyBR algorithms require less computing time than other algorithms tested, including the direct
or analytical method. Table 2 displays the measured turnaround time for each case, along with the number of iterations and the relative reconstruction error norms for each method. Compared to the direct method with fixed $\lambda$, hybrid methods require less time to compute the estimated $CO_2$ fluxes. Moreover, since the regularization parameter can be selected automatically, genHyBR methods can obtain results with smaller reconstruction errors.

Finally, we demonstrate the ability to perform UQ for AIM. In the last columns of Table 2, we provide the times needed to
compute uncertainties. We remark that the additional time and the difference in the number of iterations can be attributed to the need to perform reorthogonalization of the Krylov basis vectors, which is not as critical for obtaining the MAP estimate. Nevertheless, the additional time remains modest, given the size of the problem and the ability to obtain solution variance

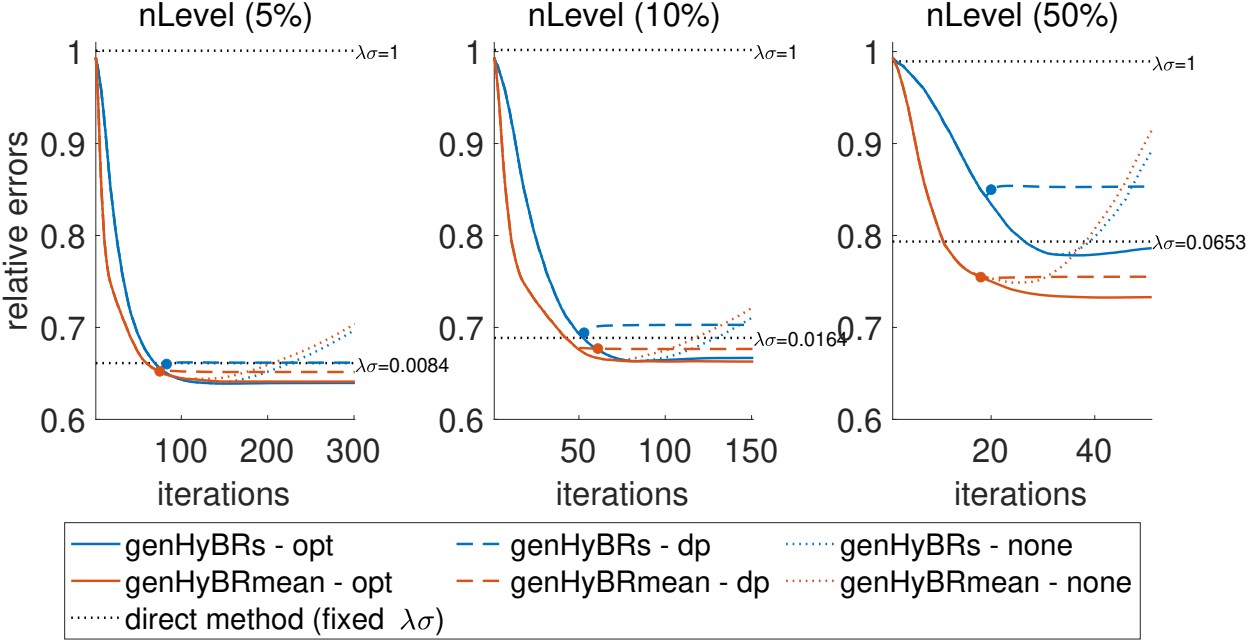

**Figure 3.** 6 weeks case study: We provide relative reconstruction error norms per iteration of `genHyBRs` and `genHyBRmean` with 5%, 10%, and 50% noise levels. We compare results for the optimal regularization parameter, the automatically selected DP parameter, and $\lambda = 0$. Relative reconstruction error norms for the direct method are provided for two fixed values of $\lambda\sigma$. The filled circle indicates the stopping iteration for the genHyBR methods with DP.



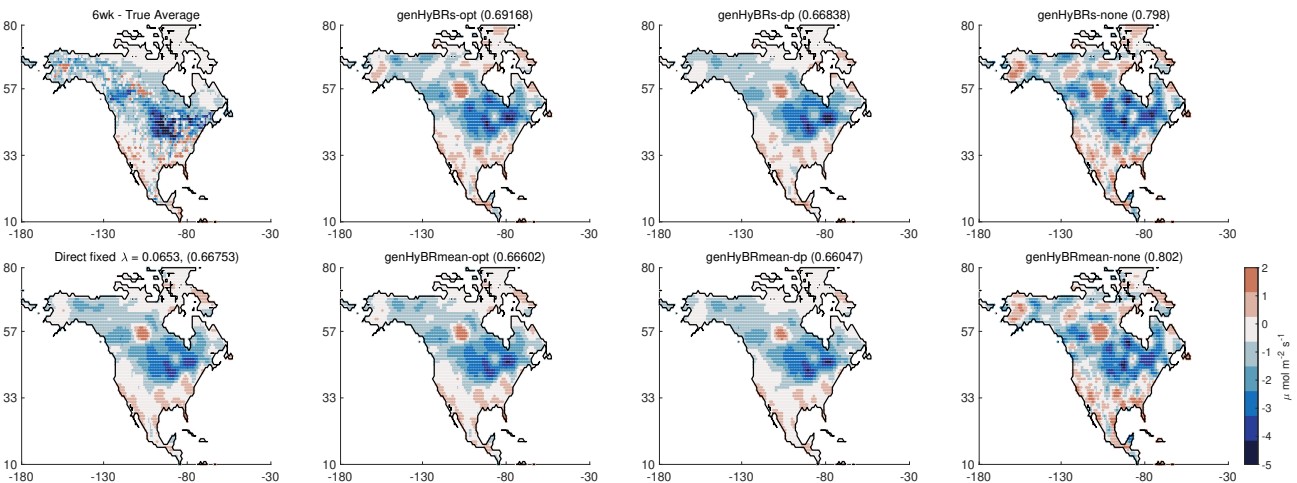

**Figure 4.** 6 weeks case study: Reconstructed fluxes, averaged over 6 weeks, are provided for `genHyBRs` and `genHyBRmean` for various parameter choices. The true average fluxes and the reconstruction using a direct method with the optimal regularization parameter computed from `genHyBRmean` are provided for comparison. These results correspond to 50% noise level and relative error norms of average fluxes over 6 weeks are provided in the titles.

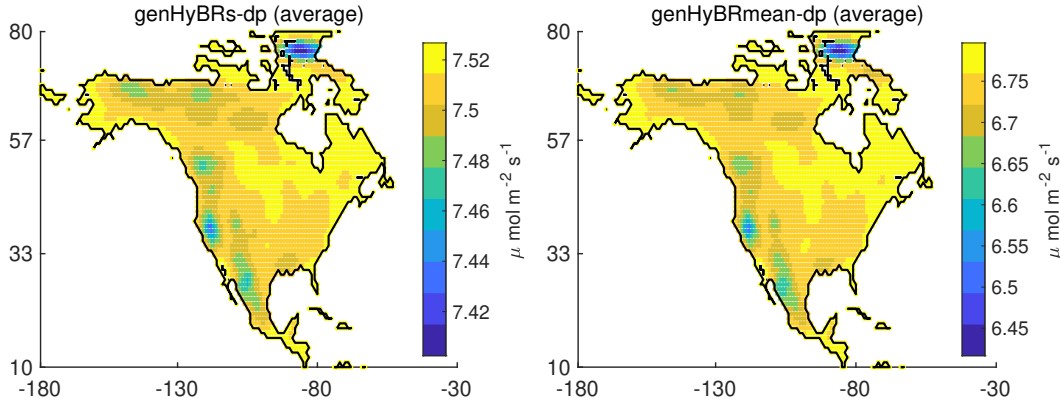

**Figure 5.** 6 weeks case study: For 50% noise level, the average posterior standard deviation over 6 weeks for both the fixed mean and the unknown mean case. DP was used to select the regularization parameter.

estimates. The estimated posterior variance is also similar across both methods (`genHyBRs` and `genHyBRmean`). Figure 5 shows the estimated averages of the posterior variance over 6 weeks for both algorithms.





| Noise Level | Methods | Selection of $\lambda\sigma$ | recons | | | recons + uncert | | |
|---|---|---|---|---|---|---|---|---|
| | | | Iter. | Time(s) | $\Delta s$ | Iter. | Time(s) | $\Delta s$ |
| 5% | direct | 0.0084 | - | 8,714 | 0.6612 | - | - | - |
| | genHyBRs-dp | 0.0137 | 99 | **4,450** | 0.6617 | 83 | 10,114 | 0.6617 |
| | genHyBRmean-dp | 0.0104 | 102 | **3,836** | 0.6515 | 75 | 8,025 | 0.6515 |
| 10% | direct | 0.0164 | - | 8,268 | 0.6886 | - | - | - |
| | genHyBRs-dp | 0.0185 | 67 | **3,125** | 0.7028 | 53 | 4,667 | 0.7028 |
| | genHyBRmean-dp | 0.0308 | 60 | **2,334** | 0.6766 | 61 | 5,572 | 0.6766 |
| 50% | direct | 0.0653 | - | 8,765 | 0.7934 | - | - | - |
| | genHyBRs-dp | 0.0750 | 20 | **922** | 0.8531 | 20 | 1,200 | 0.8532 |
| | genHyBRmean-dp | 0.0833 | 19 | **773** | 0.7551 | 18 | 981 | 0.7552 |

**Table 2.** 6 weeks case study: For various noise levels, we provide comparisons of `genHyBRs` and `genHyBRmean` (with DP selected regularization parameter) to standard direct and iterative methods (with fixed regularization parameter). We provide the number of iterations, the CPU timing in seconds, and the relative reconstruction error norms for the computed spatiotemporal fluxes denoted by $\Delta s$. Note that when computing the reconstructions and approximating the posterior variance, additional reorthogonalization is performed which explains the slight difference in the number of iterations and run time.

We finalize this section with some results for the 1 year case study. Since this case study has approximately 9 times the number of unknown $CO_2$ fluxes and 5 times the number of observations compared to the 6 weeks case study, iterative methods are computationally more appealing than direct methods for obtaining reconstructions. The previous case study already explored the behavior of the algorithms at different noise levels, so here we only consider the 50% noise level, which corresponds to $\sigma = 0.4076$. Figure 6 provides relative reconstruction error norms for `genHyBRs` and `genHyBRmean`. With the regulariza-

tion parameter automatically selected using DP, both methods result in reconstructions with relative errors smaller than 0.85. Since it is difficult to show spatiotemporal flux reconstructions over the entire year, we provide the annual average of the reconstructed $CO_2$ fluxes in Figure 7. Compared to the reconstructions in Figure 4, these average maps are much smoother. This inability to resolve fine details can be attributed to the significantly fewer observations compared to the number of unknowns in this case study. Nevertheless, these results show that the algorithms described in this study can be scaled to very large inverse

problems – problems where the direct method is either computationally prohibitive or time consuming.





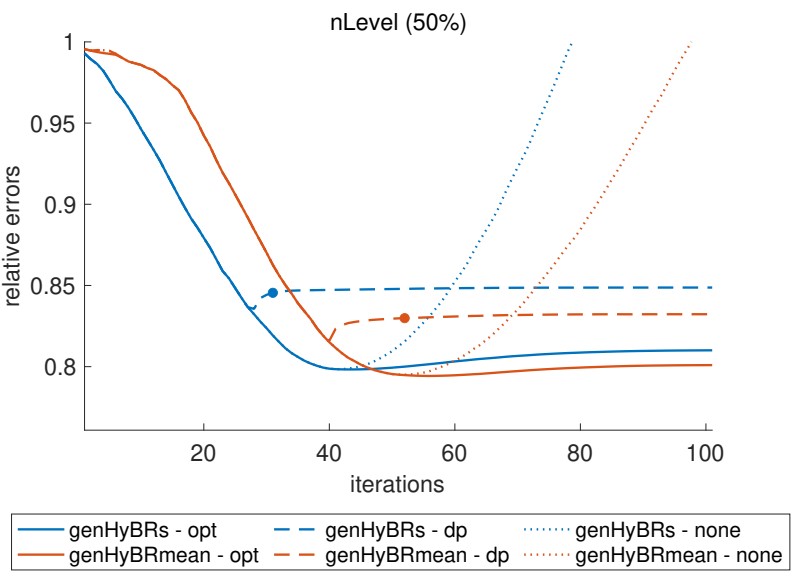

**Figure 6.** 1 year case study: For 50% noise level, we provide relative reconstruction error norms per iteration of `genHyBRs` and `genHyBRmean` and compare results for the optimal regularization parameter, the automatically selected DP parameter, and $\lambda = 0$.

## 5 Conclusions

This article describes a mathematically advanced iterative method for AIM with large datasets. Specifically, we discuss generalized hybrid methods for inverse models with a fixed prior mean (e.g., Bayesian synthesis inverse modeling) and an unknown prior mean (e.g., geostatistical inverse modeling). We also describe a means of obtaining posterior variance estimates at very
little additional computational cost. Compared to standard inverse modeling procedures (e.g., direct and iterative methods), genHyBR methods are computationally cheaper and exhibit faster convergence. One of the main advantages of genHyBR methods is the ability to efficiently and adaptively estimate the regularization parameter during the inversion process, and we described various regularization parameter estimation methods for Tikhonov regularization. Numerical experiments for case studies for 6 weeks and 1 year demonstrate that genHyBR methods provide an efficient, flexible, robust, and automatic ap-
proach for AIM with very large spatiotemporal fluxes. Furthermore, since these methods only require forward and adjoint model evaluations, these methods can be paired with different types of atmospheric transport models.

*Code availability.* The Matlab codes for the 6 weeks case study that were used to generate the results in Section 4 are available at https://doi.org/10.5281/zenodo.5772660.



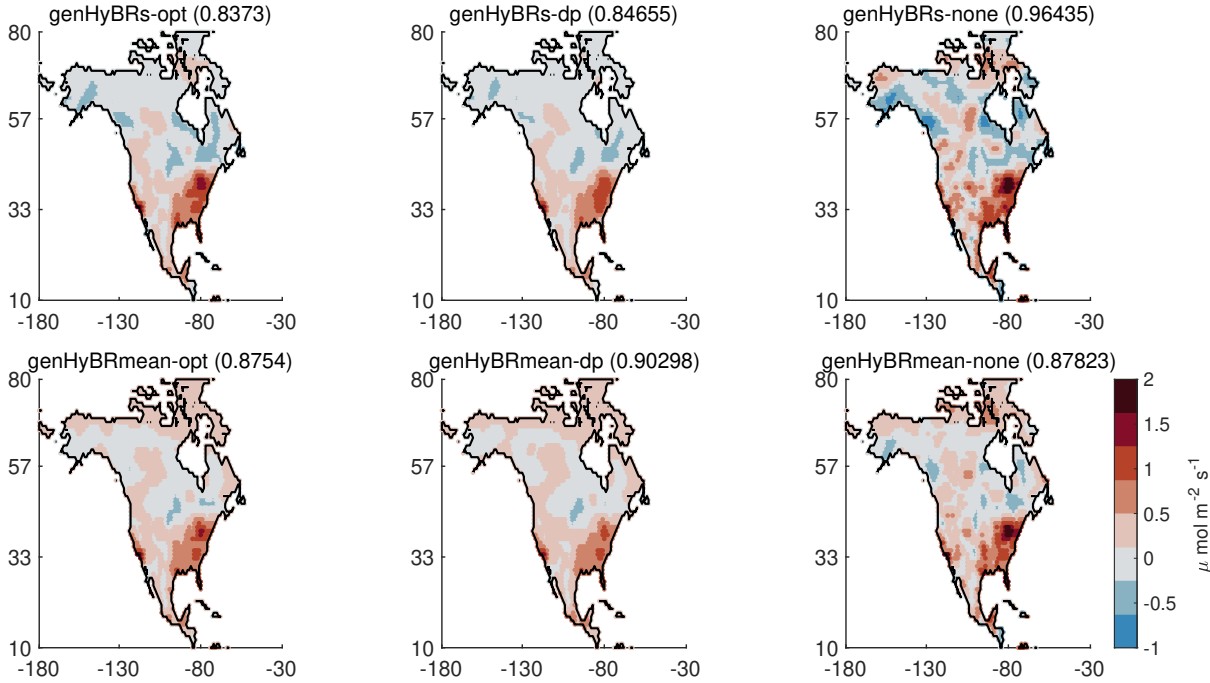

**Figure 7.** 1 year case study: Reconstructed fluxes, averaged over 1 year, are provided for `genHyBRs` and `genHyBRmean` for various parameter choices. These results correspond to 50% noise level and relative error norms of average fluxes over 1 year are provided in the titles.

## Appendix A: Regularization parameter estimation methods for genHyBR

One of the main advantages of hybrid projection methods is the ability to adaptively and automatically estimate the regularization parameter during the iterative process. We described the discrepancy principle (DP), but other common parameter estimation techniques in the context of hybrid projection methods include the generalized cross validation (GCV) method and the weighted-GCV (WGCV) method (Chung et al., 2008; Renaut et al., 2017). A summary of methods with respective functions used to compute the parameters based on the original problem and the projected problem are summarized in Table A1, where for simplicity we have assumed that $\mathbf{R} = \sigma^2 \mathbf{I}$. We have used the notation $\mathbf{B}_{k,\lambda}^\dagger = (\mathbf{B}_k^\top \mathbf{B}_k + \lambda^2 \mathbf{I})^{-1} \mathbf{B}_k^\top$ for given $\lambda > 0$ and $\mathbf{y}_{k,\lambda}$ is the solution to the projected problem (14).

More specifically, DP selects the largest parameter value $\lambda$ for which $\mathcal{D}_{\mathrm{full}}(\lambda) \le \tau m \sigma^2$, where $\tau \ge 1$ is a user-defined parameter. Note that $m\sigma^2$ is the expected value of $\|\boldsymbol{\epsilon}\|_{\mathbf{R}^{-1}}^2$. For the projected problem, we choose the largest $\lambda$ such that $\mathcal{D}_{\mathrm{proj}}(\lambda) \le \tau m \sigma^2$. The WGCV method selects $\lambda$ by minimizing the objective function $\mathcal{G}_{\mathrm{full}}(\lambda; \omega)$. Note that if $\omega = 1$ then WGCV becomes GCV. In the projected problem, we minimize $\mathcal{G}_{\mathrm{proj}}(\lambda; \omega)$ at each iteration. The parameter $\omega$ can be chosen





| Methods | Original problem (11) | Projected Problem (14) |
|---|---|---|
| DP | $\mathcal{D}_{\text{full}}(\lambda) = \|\mathbf{H}s_\lambda - z\|^2_{\mathbf{R}^{-1}}$ | $\mathcal{D}_{\text{proj}}(\lambda) = \|\mathbf{B}_k y_{k,\lambda} - \delta_1 e_1\|^2_2$ |
| GCV | $\mathcal{G}_{\text{full}}(\lambda;\omega) = \dfrac{m\|\mathbf{H}x_\lambda - b\|^2_{\mathbf{R}^{-1}}}{(\text{tr}(\mathbf{I} - \omega\mathbf{H}\mathbf{H}^\dagger_\lambda))^2}$ | $\mathcal{G}_{\text{proj}}(\lambda;\omega) = \dfrac{k\|\mathbf{B}_k y_{k,\lambda} - \delta_1 e_1\|^2_2}{(\text{tr}(\mathbf{I} - \omega\mathbf{B}_k\mathbf{B}^\dagger_{k,\lambda}))^2}$ |

**Table A1.** Regularization parameter selection methods for use within genHyBR methods.

automatically, as described in (Chung et al., 2008; Renaut et al., 2017). Note that the DP approach requires a priori knowledge of the noise variance $\sigma^2$, whereas the GCV approaches do not require prior knowledge about the noise level.

### Appendix B: Extension to unknown mean: Hierarchical Bayes

In this section, we provide details for the derivation of genHyBR methods for AIM with unknown mean. We begin in Appendix

B1 with the problem reformulation to simultaneously estimate the unknown fluxes and the covariate parameters. Then in B2 we provide the details of the genHyBR approach for the unknown mean case, which closely follows the derivation in Section 3.1.1. Finally, in Appendix B3 we show how to approximate the posterior variance using the generalized Golub-Kahan bidiagonalization.

### B1    Reformulation for simultaneous estimation

In order to apply genHyBR methods to the unknown mean estimation problem, we first reformulation the MAP estimate from (7) to (18) as follows. For the data fit term, consider

$$
\begin{aligned}
\frac{1}{2}\|\mathbf{H}s - z\|^2_{\mathbf{R}^{-1}} &= \frac{1}{2}\left\|\begin{bmatrix}\mathbf{H} & \mathbf{0}\end{bmatrix}\begin{bmatrix}s \\ \boldsymbol{\beta}\end{bmatrix} - z\right\|^2_{\mathbf{R}^{-1}} \\
&= \frac{1}{2}\left\|\mathbf{K}\begin{bmatrix}\widetilde{s} + \mathbf{X}\boldsymbol{\mu}_\beta \\ \widetilde{\boldsymbol{\beta}} + \boldsymbol{\mu}_\beta\end{bmatrix} - z\right\|^2_{\mathbf{R}^{-1}} \\
&= \frac{1}{2}\left\|\mathbf{K}\begin{bmatrix}\widetilde{s} \\ \widetilde{\boldsymbol{\beta}}\end{bmatrix} + \mathbf{H}\mathbf{X}\boldsymbol{\mu}_\beta - z\right\|^2_{\mathbf{R}^{-1}} \\
&= \frac{1}{2}\left\|\mathbf{K}\begin{bmatrix}\widetilde{s} \\ \widetilde{\boldsymbol{\beta}}\end{bmatrix} - \widetilde{z}\right\|^2_{\mathbf{R}^{-1}}
\end{aligned}
$$





and for the regularization terms in (7), and we have

$$
\begin{aligned}
\frac{\lambda^2}{2}\|s - \mathbf{X}\beta\|_{\mathbf{Q}_s^{-1}}^2 + \frac{\lambda_\beta^2}{2}\|\beta - \mu_\beta\|_{\mathbf{Q}_\beta^{-1}}^2 &= \frac{\lambda^2}{2}\|\widetilde{s} - \mathbf{X}\widetilde{\beta}\|_{\mathbf{Q}_s^{-1}}^2 + \frac{\lambda_\beta^2}{2}\|\widetilde{\beta}\|_{\mathbf{Q}_\beta^{-1}}^2 \\
&= \frac{1}{2}(\lambda^2\widetilde{s}^\top \mathbf{Q}_s^{-1}\widetilde{s} - 2\lambda^2\widetilde{s}^\top \mathbf{Q}_s^{-1}\mathbf{X}\widetilde{\beta} + \widetilde{\beta}^\top(\lambda^2\mathbf{X}^\top\mathbf{Q}_s^{-1}\mathbf{X} + \lambda_\beta^2\mathbf{Q}_\beta^{-1})\widetilde{\beta}) \\
&= \frac{1}{2}(\lambda^2\widetilde{s}^\top \mathbf{Q}_s^{-1}\widetilde{s} - 2\lambda^2\widetilde{s}^\top \mathbf{Q}_s^{-1}\mathbf{X}\widetilde{\beta} + \widetilde{\beta}^\top(\lambda^2\mathbf{X}^\top\mathbf{Q}_s^{-1}\mathbf{X} + (\alpha\lambda)^2\mathbf{Q}_\beta^{-1})\widetilde{\beta}) \\
&= \frac{1}{2}\begin{bmatrix}\widetilde{s}^\top & \widetilde{\beta}^\top\end{bmatrix}\begin{bmatrix}\lambda^2\mathbf{Q}_s^{-1} & -\lambda^2\mathbf{Q}_s^{-1}\mathbf{X} \\ -\lambda^2\mathbf{X}^\top\mathbf{Q}_s^{-1} & \lambda^2\mathbf{X}^\top\mathbf{Q}_s^{-1}\mathbf{X} + (\alpha\lambda)^2\mathbf{Q}_\beta^{-1}\end{bmatrix}\begin{bmatrix}\widetilde{s} \\ \widetilde{\beta}\end{bmatrix} \\
&= \frac{\lambda^2}{2}\begin{bmatrix}\widetilde{s}^\top & \widetilde{\beta}^\top\end{bmatrix}\underbrace{\begin{bmatrix}\mathbf{Q}_s^{-1} & -\mathbf{Q}_s^{-1}\mathbf{X} \\ -\mathbf{X}^\top\mathbf{Q}_s^{-1} & \mathbf{X}^\top\mathbf{Q}_s^{-1}\mathbf{X} + \alpha^2\mathbf{Q}_\beta^{-1}\end{bmatrix}}_{=\mathbf{Q}^{-1}}\begin{bmatrix}\widetilde{s} \\ \widetilde{\beta}\end{bmatrix} \\
&= \frac{\lambda^2}{2}\|\widetilde{\gamma}\|_{\mathbf{Q}^{-1}}^2.
\end{aligned}
$$

We identify $\widetilde{\gamma} = [\widetilde{s}^\top, \widetilde{\beta}^\top]^\top$ and this completes the derivation of (18).

Next we provide the derivation of the augmented prior covariance matrix (19). Since we need $\mathbf{Q}$ and not $\mathbf{Q}^{-1}$ in genHyBR, we use the formula for the inverse of a $2 \times 2$ block matrix

$$
\begin{bmatrix}\mathbf{A} & \mathbf{B} \\ \mathbf{C} & \mathbf{D}\end{bmatrix}^{-1} = \begin{bmatrix}(\mathbf{A} - \mathbf{B}\mathbf{D}^{-1}\mathbf{C})^{-1} & -(\mathbf{A} - \mathbf{B}\mathbf{D}^{-1}\mathbf{C})^{-1}\mathbf{B}\mathbf{D}^{-1} \\ -\mathbf{D}^{-1}\mathbf{C}(\mathbf{A} - \mathbf{B}\mathbf{D}^{-1}\mathbf{C})^{-1} & \mathbf{D}^{-1}\mathbf{C}(\mathbf{A} - \mathbf{B}\mathbf{D}^{-1}\mathbf{C})^{-1}\mathbf{B}\mathbf{D}^{-1} + \mathbf{D}^{-1}\end{bmatrix}
$$

which is defined if $\mathbf{D}$ and $\mathbf{A} - \mathbf{B}\mathbf{D}^{-1}\mathbf{C}$ are invertible. Using the above formula, we get

$\mathbf{Q} = \begin{bmatrix}\mathbf{Q}_s^{-1} & -\mathbf{Q}_s^{-1}\mathbf{X} \\ -\mathbf{X}^\top\mathbf{Q}_s^{-1} & \mathbf{X}^\top\mathbf{Q}_s^{-1}\mathbf{X} + \alpha^2\mathbf{Q}_\beta^{-1}\end{bmatrix}^{-1} = \begin{bmatrix}\mathbf{Q}_s + \frac{1}{\alpha^2}\mathbf{X}\mathbf{Q}_\beta\mathbf{X}^\top & \frac{1}{\alpha^2}\mathbf{X}\mathbf{Q}_\beta \\ \frac{1}{\alpha^2}\mathbf{Q}_\beta\mathbf{X}^\top & \frac{1}{\alpha^2}\mathbf{Q}_\beta\end{bmatrix}.$

This simplifies to

$$
\mathbf{Q} = \begin{bmatrix}\mathbf{Q}_s & \mathbf{0} \\ \mathbf{0} & \mathbf{0}\end{bmatrix} + \frac{1}{\alpha^2}\begin{bmatrix}\mathbf{X} \\ \mathbf{I}\end{bmatrix}\mathbf{Q}_\beta\begin{bmatrix}\mathbf{X}^\top & \mathbf{I}\end{bmatrix},
$$

which completes the derivation of (19).

## B2   genHyBR approach for AIM with unknown mean

In this subsection, we derive the genHyBR approach for the unknown mean case. We initialize $\delta_1^K = \|\widetilde{z}\|_{\mathbf{R}^{-1}}$, $u_1^K = \widetilde{z}/\delta_1^K$ and $\gamma_1^K v_1^K = \mathbf{K}^\top\mathbf{R}^{-1}u_1^K$, then the $k$th iteration of the generalized Golub-Kahan bidiagonalization procedure generates vectors $u_{k+1}^K$ and $v_{k+1}^K$ such that

$$
\begin{aligned}
\gamma_{k+1}^K u_{k+1}^K &= \mathbf{K}\mathbf{Q}v_k^K - \gamma_k^K u_k^K \\
\delta_{k+1}^K v_{k+1}^K &= \mathbf{K}^\top\mathbf{R}^{-1}u_{k+1}^K - \delta_{k+1}^K v_k^K
\end{aligned}
$$





where scalars $\gamma_k^K, \delta_k^K \geq 0$ are computed such that $\|\boldsymbol{u}_k^K\|_{\mathbf{R}^{-1}} = \|\boldsymbol{v}_k^K\|_{\mathbf{Q}} = 1$. At the end of $k$ iterations, we have

$$
\mathbf{B}_k^K \equiv \begin{bmatrix} \gamma_1^K & & & & \\ \delta_2^K & \gamma_2^K & & & \\ & \delta_3^K & \ddots & & \\ & & \ddots & \gamma_k^K & \\ & & & \delta_{k+1}^K & \end{bmatrix}, \quad \mathbf{U}_{k+1}^K \equiv \begin{bmatrix} \boldsymbol{u}_1^K, \dots, \boldsymbol{u}_{k+1}^K \end{bmatrix}, \quad \text{and} \quad \mathbf{V}_k^K \equiv \begin{bmatrix} \boldsymbol{v}_1^K, \dots, \boldsymbol{v}_k^K \end{bmatrix}.
$$

The above matrices satisfy the following relations

$$
\begin{aligned}
\mathbf{U}_{k+1}^K \delta_1^K \boldsymbol{e}_1 &= \widetilde{\boldsymbol{z}} \\
\mathbf{KQV}_k^K &= \mathbf{U}_{k+1}^K \mathbf{B}_k^K \\
\mathbf{K}^\top \mathbf{R}^{-1} \mathbf{U}_{k+1}^K &= \mathbf{V}_k^K (\mathbf{B}_k^K)^\top + \gamma_{k+1}^K \boldsymbol{v}_{k+1}^K \boldsymbol{e}_{k+1}^\top.
\end{aligned}
\tag{B1}
$$

Also, matrices $\mathbf{U}_{k+1}^K$ and $\mathbf{V}_k^K$ satisfy the following orthogonality conditions:

$\quad (\mathbf{U}_{k+1}^K)^\top \mathbf{R}^{-1} \mathbf{U}_{k+1}^K = \mathbf{I}_{k+1} \quad \text{and} \quad (\mathbf{V}_k^K)^\top \mathbf{Q} \mathbf{V}_k^K = \mathbf{I}_k.$ (B2)

The columns of the matrix $\mathbf{V}_k^K$ form a basis for the Krylov subspace $\mathcal{K}_k(\mathbf{K}^\top \mathbf{R}^{-1} \mathbf{KQ}, \mathbf{K}^\top \mathbf{R}^{-1} \widetilde{\boldsymbol{z}})$, which we use to search for approximate solutions.

To obtain the approximate solution, we solve the least-squares problem

$$
\min_{\boldsymbol{y} \in \mathbb{R}^k} \|\mathbf{B}_k^K \boldsymbol{y} - \delta_1^K \boldsymbol{e}_1\|_2^2 + \lambda^2 \|\boldsymbol{y}\|_2^2,
\tag{B3}
$$

to obtain the optimizer $\boldsymbol{y}_{k,\lambda}^K$ and to compute the approximate solution $\boldsymbol{\gamma}_{k,\lambda} = \mathbf{Q} \mathbf{V}_k^K \boldsymbol{y}_{k,\lambda}^K$. We can extract the approximations $\widetilde{\boldsymbol{s}}_{k,\lambda}$ and $\widetilde{\boldsymbol{\beta}}_{k,\lambda}$ as $\boldsymbol{\gamma}_{k,\lambda} = \begin{bmatrix} \widetilde{\boldsymbol{s}}_{k,\lambda} \\ \widetilde{\boldsymbol{\beta}}_{k,\lambda} \end{bmatrix}$. To estimate the regularization parameter, we can adapt the techniques in Section 3.1.2; for example, using the Discrepancy principle, we pick a regularization parameter $\lambda$ such that

$$
\mathcal{D}_{\text{proj}}^K(\lambda) = \|\mathbf{B}_k^K \boldsymbol{y}_{k,\lambda}^K - \delta_1^K \boldsymbol{e}_1\|_2^2 \leq \tau m,
$$

where $\tau \geq 1$ is a user-defined parameter and $m$ is the expected value of $\|\boldsymbol{\epsilon}\|_{\mathbf{R}^{-1}}^2$. Other parameter selection techniques such 500 as GCV and WGCV can also be adapted to the unknown mean case with similar expressions as in Table A1 but we omit the details.

## B3 Approximation to the posterior variance

We propose an approximation to the posterior covariance matrix $\mathbf{\Gamma}_{\text{post}}$ corresponding to the posterior distribution $\pi(\boldsymbol{s}, \boldsymbol{\beta}|\boldsymbol{z})$ in (6). First notice that from (19) and (8), we get the following expression of the posterior covariance matrix

$\mathbf{\Gamma}_{\text{post}} = \left( \lambda^2 \mathbf{Q}^{-1} + \mathbf{K}^\top \mathbf{R}^{-1} \mathbf{K} \right)^{-1} = \mathbf{Q} \left( \lambda^2 \mathbf{Q} + \mathbf{Q} (\mathbf{K}^\top \mathbf{R}^{-1} \mathbf{K}) \mathbf{Q} \right)^{-1} \mathbf{Q},$ (B4)



where the last expression is obtained by factoring out $\mathbf{Q}$. Assume that $k$ iterations of the generalized Golub-Kahan bidiagonalization have been performed to solve (18). Let $(\mathbf{B}_k^K)^\top \mathbf{B}_k^K = \mathbf{W}_k^K \mathbf{\Theta}_k^K (\mathbf{W}_k^K)^\top$ be an eigenvalue decomposition with eigenvalues $\theta_1^K, \ldots, \theta_k^K$ and let $\mathbf{Z}_k^K = \mathbf{Q}\mathbf{V}_k^K\mathbf{W}_k^K$. Then consider the following low-rank approximation,

$$\mathbf{Q}(\mathbf{K}^\top \mathbf{R}^{-1}\mathbf{K})\mathbf{Q} \approx \mathbf{Q}(\mathbf{V}_k^K(\mathbf{B}_k^K)^\top \mathbf{B}_k^K \mathbf{V}_k^\top)\mathbf{Q} = \mathbf{Z}_k^K \mathbf{\Theta}_k^K (\mathbf{Z}_k^K)^\top. \tag{B5}$$

Using (B5) in (B4), we can define an approximate covariance matrix as

$$\widetilde{\mathbf{\Gamma}}_{\text{post}} = \mathbf{Q}(\lambda^2 \mathbf{Q} + \mathbf{Z}_k^K \mathbf{\Theta}_k^K (\mathbf{Z}_k^K)^\top)^{-1}\mathbf{Q} = \lambda^{-2}\mathbf{Q} - \mathbf{Z}_k^K \mathbf{\Delta}_k^K (\mathbf{Z}_k^K)^\top, \tag{B6}$$

where $\mathbf{\Delta}_k^K$ is diagonal matrix,

$$\mathbf{\Delta}_k^K \equiv \lambda^{-2}\begin{bmatrix} \frac{\theta_1^K}{\theta_1^K+\lambda^2} & & \\ & \ddots & \\ & & \frac{\theta_k^K}{\theta_k^K+\lambda^2} \end{bmatrix} \in \mathbb{R}^{k\times k}.$$

The last expression was obtained using the Woodbury formula.

Therefore, we have an efficient representation of the approximate posterior covariance matrix as a low-rank perturbation of the prior covariance matrix $\mathbf{Q}$. It is important to note that as with the prior covariance matrix $\mathbf{Q}$, we do not need to store $\widetilde{\mathbf{\Gamma}}_{\text{post}}$ explicitly. More precisely, in addition to storing the information required for storing $\mathbf{Q}$, we only need to store $nk+k$ additional entries corresponding to the matrices $\mathbf{Z}_k^K$ and $\mathbf{\Delta}_k^K$. Furthermore, the error in the low-rank approximation can be analyzed using similar techniques as in (Saibaba et al., 2020). Similar to the approach described in Section 3.2, the posterior variance,

which corresponds to the diagonal entries of $\mathbf{\Gamma}_{\text{post}}$, can be approximated using the diagonal entries of (B6). First, note the diagonals of $\mathbf{Q}$ are obtained from the diagonals of the block matrices $\mathbf{Q}_s + \alpha^{-2}\mathbf{X}\mathbf{Q}_\beta\mathbf{X}^\top$ and $\alpha^{-2}\mathbf{Q}_\beta$. The diagonals of $\mathbf{Q}_s$ and $\mathbf{Q}_\beta$ are typically known analytically. The diagonals of $\mathbf{X}\mathbf{Q}_s\mathbf{X}^\top$ and $\mathbf{Z}_k^K \mathbf{\Delta}_k^K (\mathbf{Z}_k^K)^\top$ are easy to compute in $\mathcal{O}((n+p)k^2)$ flops since they are low-rank matrices. Therefore, given the approximate representation of the covariance matrix (B6), we can estimate the posterior variance (i.e., the diagonals of the posterior covariance). A complete description of the method is given

in Algorithm 3.

*Author contributions.* JC, SMM and AKS designed the study. TC conducted the numerical simulations. All authors contributed to the writing and presentation of results.

*Competing interests.* The authors declare that they have no conflict of interest.

*Acknowledgements.* This work was partially supported by the National Science Foundation ATD program under grants DMS-2026841,
2026830, and 2026835. This material was also based upon work partially supported by the National Science Foundation under Grant DMS-1638521 to the Statistical and Applied Mathematical Sciences Institute. The OCO-2 CarbonTracker-Lagrange footprint library was produced




---

**Algorithm 3** AIM with unknown mean—`genHyBRmean` with UQ

---

**Require:** Matrices $\mathbf{K}$, $\mathbf{R}$ and $\mathbf{Q}$, and vector $\widetilde{z}$.

1: {/∗ Compute MAP estimate ∗/}

2: initialize $\boldsymbol{u}_1^K = \widetilde{z}/\|\widetilde{z}\|_{\mathbf{R}^{-1}}$

3: **for** $j = 1, \ldots, k$ **do**

4:     one iteration of generalized Golub-Kahan bidiagonalization to obtain $\mathbf{B}_j^K$, $\mathbf{U}_{j+1}^K$, and $\mathbf{V}_j^K$

5:     estimate regularization parameter $\lambda$ and compute $\boldsymbol{y}_{j,\lambda}^K$ by solving (B3)

6: **end for**

7: compute $\begin{bmatrix} \widetilde{\boldsymbol{s}}_{k,\lambda} \\ \widetilde{\boldsymbol{\beta}}_{k,\lambda} \end{bmatrix} = \mathbf{Q}\mathbf{V}_k^K \boldsymbol{y}_{k,\lambda}^K$ and $\boldsymbol{\gamma}_{k,\lambda} = \begin{bmatrix} \widetilde{\boldsymbol{s}}_{k,\lambda} + \mathbf{X}\boldsymbol{\mu}_\beta \\ \widetilde{\boldsymbol{\beta}}_{k,\lambda} + \boldsymbol{\mu}_\beta \end{bmatrix}$

8: {/∗ Compute the approximation to the posterior variance ∗/}

9: compute the eigendecomposition $(\mathbf{B}_k^K)^\top \mathbf{B}_k^K = \mathbf{W}_k^K \boldsymbol{\Theta}_k^K (\mathbf{W}_k^K)^\top$

10: compute $\mathbf{Z}_k^K = \mathbf{Q}\mathbf{V}_k^K \mathbf{W}_k^K$ and $\boldsymbol{\Delta}_k^K = \mathrm{diag}(\frac{\theta_1^K}{\theta_1^K + \lambda^2}, \ldots, \frac{\theta_k^K}{\theta_k^K + \lambda^2})$

11: compute $\boldsymbol{d}_{\mathrm{LR}} = \mathrm{LowRankDiag}(\mathbf{Z}_k^K \boldsymbol{\Delta}_k^K, \mathbf{Z}_k^K)$ using Algorithm 2

12: compute $[\boldsymbol{d}_\beta] = \mathrm{LowRankDiag}(\mathbf{X}\mathbf{Q}_\beta, \mathbf{X})$, $\boldsymbol{d}_{\mathbf{Q}} = [\mathrm{diag}(\mathbf{Q}_s) + \alpha^{-2}\boldsymbol{d}_\beta; \alpha^{-2}\mathrm{diag}(\mathbf{Q}_\beta)]$

13: estimate posterior variance $\boldsymbol{d}_{k,\lambda} = \lambda^{-2}\boldsymbol{d}_{\mathbf{Q}} - \boldsymbol{d}_{\mathrm{LR}}$

14: **return** MAP estimate $\boldsymbol{\gamma}_{k,\lambda}$ and variance estimate $\boldsymbol{d}_{k,\lambda}$

---

by NOAA/GML and AER Inc with support from NASA Carbon Monitoring System project "Andrews (CMS 2014) Regional Inverse Modeling in North and South America for the NASA Carbon Monitoring System." We especially thank Arlyn Andrews, Michael Trudeau, and Marikate Mountain for generating and assisting with the footprint library. Any opinions, findings, and conclusions or recommendations expressed in this material are those of the author(s) and do not necessarily reflect the views of the National Science Foundation.





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
