# Peer review of "Computationally efficient methods for large-scale atmospheric inverse modeling"

_Geoscientific Model Development, 2021_

## Referee Comment (RC1)

**Overview**

This work presents two generalized hybrid methods for atmospheric inversion modeling schemes, one involving a fixed prior mean and another with an unknown prior mean. These methods offer greater computational efficiency during the inversion process and approximate posterior covariance matrices that may otherwise be unfeasible to calculate. Results were presented in the context of a continental inversion scheme using synthetic CO2 observations from the OCO-2 instrument. Although this manuscript is mathematically dense, it is well written and informative. The given detail of the Bayesian inversion scheme may narrow the readership of this article to those specializing in this type of mathematics while being more challenging to implement by those who have only an operational knowledge of inversions. Nonetheless, this work provides a timely improvement to the atmospheric inversion modeling process as the number of space-based CO2 observing platforms is set to increase. I recommend that this article be published once the following comments have been addressed.

**General Comments**

The introduction is clear and well written; however, additional motivation would strengthen this work. Currently, the introduction poses the content as an "interesting math problem" but fails to answer the question of *why are inversions so important that they need to be done faster?* Since this publication was submitted to GMD, the physical implications of this work should also be mentioned. Observation systems are increasing in number, allowing scientists to better constrain anthropogenic influences on the climate. To mitigate these influences, rapid monitoring, reporting, and verification of emissions (anthropogenic and otherwise) is needed to ensure cities, regions, and nations are working to reduce them. Ultimately, the work presented in this manuscript will assist with this challenge, making it more than just "interesting math". These points should be reiterated in the conclusion.

In line 22, it is stated that "the number of greenhouse gases and air pollution measurements has greatly expanded in the past decade, enabling investigations of surface fluxes across larger regions, longer time periods, and/or at finer spatial and temporal detail." However, the only example of a ground-based network given was NOAA's GML. Several other local/regional ground-based monitoring systems exist and could be offered to readers as additional examples. Salt Lake City's UUCON, Indianapolis INFLUX, UC Berkeley's BEACON. (Some of these networks may provide data to NOAA's GML data archive but their local/regional focus is worth noting.)

Approximate posterior covariance matrices, $\widetilde{Q}_{\text{post}}$ and $\tilde{\Gamma}_{\text{post}}$ are given in line 264 and Equation B6 respectively. For the effectiveness of these approximation methods, the reader is referred to a citation: Saibaba et al., 2020. A few sentences within this manuscript describing results from Saibaba et al., the effectiveness of the approximations, limitations, etc. would benefit the curious reader.

The two case studies reported in this work made use of pseudo-observations from NASA's OCO-2 instrument to estimate CO2 fluxes at 3-hour intervals and 1º x 1º spatial resolution. It is unclear from the text alone how OCO-2 soundings were incorporated into the inversion scheme. Generally, OCO-2 soundings are densely spaced (~2-3km apart) and demonstrate varying spatial correlation in error. How is the assumption of spatially independent errors ($R = \sigma^2 I$) justified?

How/if soundings were spatiotemporally aggregated for this study is not clear. These details should be *briefly* included in the text while referring readers to other sources for *more* detail.

Figures 4 and 7 present the results of this work in a concise way; however, it is difficult to compare the effectiveness of so many different methods. Consider plotting the *differences* between the estimated and true fluxes. Obviously, the goal is to get as close to the true flux as possible. So, using a blue (-) to red (+) color gradient will easily show *where* the inversion is overestimating, underestimating, and effectively reproducing true emissions. Comparing gradients associated with relative differences may be easier across the various model outputs.

**Technical Specifics**
Moving the citations in line 28 to immediately follow its associated observing system instead of listing them at the end of the statement would be helpful for readers.

In line 82, it may be worth pointing out, for readers unfamiliar with this technique, that flux values from spatially explicit 2D arrays (x,y) are represented as a vector in this technique. Thus, $n$ is the number of cells in the domain of interest. i.e. – enforce how $\vec{s}$ is constructed.

Minor point: In lines 52, 139, 150, 157, 247, 254, 265, 370, papers are referenced directly in the text but they still have parentheses around them. Typically, parentheses are only included if a statement is cited without direct reference in the text. (Is this GMD formatting?)

On line 247, the sentence beginning with "Instead, we follow the approach described in…" the word 'using' is included twice, making the sentence awkward to read.

In line 291, $p$ appears to be "filler" dimensions in the matrix to ensure that matrix multiplications work out. Although this can be intuited from the mathematics, it would be beneficial to point it out in the text.

In line 343, $\sigma = 2$. It is suggested in the text that this corresponds to nlevel 100%. So, how can nlevel 50% be 0.5648 as in Table 1? Isn't the percentage of nlevel relative to this value from Miller et al., 2020? This is unclear in the text.

What are the units of $\theta_t$ and $\theta_g$ in line 370?

Apparent typo in line 373: "… sparse can…" should be "… sparse and can…"

Figure #3 needs some work. The axes and title texts need to be made smaller to better fit the in-plot labels of $\lambda\sigma$. Smaller text will allow for bigger plot areas.

---

## Referee Comment (RC2)

**1 General comments**

The authors present new iterative computational methods for performing atmospheric inversions. They treat two cases, in which the mean of the fluxes is specified directly, and in which the mean of the fluxes is unknown and is described by a parametric linear model. Unknown covariance/regularisation parameters can be estimated as part of the iterative approach at relatively low additional computational cost. As a by-product of the method, an approximation of the posterior covariance matrix may be constructed. The methods are described clearly and the authors make a contribution to the state-of-the-art for atmospheric inversions. I support publication after the following concerns are addressed.

I think the authors could devote a bit more space to reviewing other iterative methods and placing their method in that context. For example, the method appears to fall within the broad class of variational methods. A few papers for these methods are cited but it would be useful for the reader to understand a little more about the proposed method in context, and (the subject of my next comment) how it improves upon existing methods.

Connected to the previous comment, I find it hard to evaluate the relative computational benefits of the method. It is clear that the method is faster than the direct inversion. However, is it faster (e.g., takes fewer iterations) than, say, methods proposed in Miller et al. (2020)? How does the computational complexity compare? Relatedly, all the methods tested make use of the fact that the transport matrix, $\mathbf{H}$, was precomputed. That is no limitation of the method, which does not rely on this fact, but it does make it harder to understand the *total* time required for the inversion, which properly includes the precomputation of $\mathbf{H}$. It would be good for the authors to mention this, and perhaps speak a little to the computational burden of calculating $\mathbf{H}$. This is important because the authors suggest that one of primary contributions of the method is its computational efficiency.

The authors present uncertainty quantification for the fluxes but, unlike the reconstructed fluxes, there is no evaluation of whether the approximate covariance matrix is suitable. My concern would be that the low rank approximation could over estimate marginal variances and underestimate correlations (though whether this is true is not obvious to me). A second concern is that the regularisation parameter $\lambda$ is changing in every iteration (probably more so in earlier iterations), whereas I believe the theoretical results for the posterior covariance in Saibaba et al. (2020) depend on knowing $\lambda$ (though I may have misunderstood this). These concerns could be addressed empirically by evaluating the suitability of the uncertainties. One simplistic way do this would be to investigate this would be to compare the approximate posterior standard deviations to those from the direct method, which should be available for the six week case study. A second way would be to discuss in slightly more detail what the results of Saibaba et al. (2020) say about this. A third way, perhaps less feasible, would be to consider a scoring rule that takes into account the uncertainty (similar to the reconstruction error for the posterior mean). One such scoring rule could be the posterior log density (which is multivariate Gaussian) evaluated at the truth—higher density would be better. Another scoring rule option could be to consider the calibration of the uncertainty intervals for some quantity of interest such as the total flux (for example, do the 50% prediction intervals for the total flux in each time step contain the total flux 50% of the time?) These ideas are just meant as suggestions, the more general request being to explore whether the uncertainties are suitable.

**2 Specific comments**

My specific comments are:

- In Section 3.1.2, it would be good for the authors to provide a reference that specifically discusses the use of the discrepancy principle in similar problems.

- On page 14, line 358, it is stated that $\mathbf{X}$ is identical to Miller et al. (2020). But in Miller et al., $\mathbf{X}$ is described as having only one column, an intercept. Have I missed something?

**3    Technical corrections**

- At several places in-text citations are surrounded in parentheses, e.g. written as (Smith et al., 2020) instead of Smith et al. (2020). Generally I would expect to see the latter when the citation is incorporated into the sentence.

**References**

Miller, S. M., Saibaba, A. K., Trudeau, M. E., Mountain, M. E., and Andrews, A. E. (2020). Geostatistical inverse modeling with very large datasets: An example from the Orbiting Carbon Observatory 2 (OCO-2) satellite. *Geoscientific Model Development*, 13(3):1771–1785.

Saibaba, A. K., Chung, J., and Petroske, K. (2020). Efficient Krylov subspace methods for uncertainty quantification in large Bayesian linear inverse problems. *Numerical Linear Algebra with Applications*, 27(5):e2325.

---

## Author Comment (AC1)

**Computationally efficient methods for large-scale atmospheric inverse modeling**

T. Cho, J. Chung, S. M. Miller, and A. K. Saibaba

June 2, 2022

We are grateful to the reviewers for their careful reading and thoughts on our manuscript. Below, we repeat the reviewers' remarks and interleave our responses. All modifications in the revised manuscript are highlighted in **blue** and all references to equations, pages, lines, and citations correspond to the revised manuscript.

**1 Responses to Referee #1**

**Overview** This work presents two generalized hybrid methods for atmospheric inversion modeling schemes, one involving a fixed prior mean and another with an unknown prior mean. These methods offer greater computational efficiency during the inversion process and approximate posterior covariance matrices that may otherwise be unfeasible to calculate. Results were presented in the context of a continental inversion scheme using synthetic CO2 observations from the OCO-2 instrument. Although this manuscript is mathematically dense, it is well written and informative. The given detail of the Bayesian inversion scheme may narrow the readership of this article to those specializing in this type of mathematics while being more challenging to implement by those who have only an operational knowledge of inversions. Nonetheless, this work provides a timely improvement to the atmospheric inversion modeling process as the number of space-based CO2 observing platforms is set to increase. I recommend that this article be published once the following comments have been addressed.

**General Comments**

1. The introduction is clear and well written; however, additional motivation would strengthen this work. Currently, the introduction poses the content as an "interesting math problem" but fails to answer the question of why are inversions so important that they need to be done faster? Since this publication was submitted to GMD, the physical implications of this work should also be mentioned. Observation systems are increasing in number, allowing scientists to better constrain anthropogenic influences on the climate. To mitigate these influences, rapid monitoring, reporting, and verification of emissions (anthropogenic and otherwise) is needed to ensure cities, regions, and nations are working to reduce them. Ultimately, the work presented in this manuscript will assist with this challenge, making it more than just "interesting math". These points should be reiterated in the conclusion.

   We have added text to the introduction that explains this motivation. Specifically, we believe it is not only important to estimate solutions to inverse problems faster but also be able to assimilate large numbers of observations to estimate fluxes at higher spatial and temporal resolutions. Doing so will help make the most of the information available in atmospheric observations.

2. In line 22, it is stated that "the number of greenhouse gases and air pollution measurements has greatly expanded in the past decade, enabling investigations of surface fluxes across larger regions, longer

time periods, and/or at finer spatial and temporal detail." However, the only example of a ground-based network given was NOAA's GML. Several other local/regional ground-based monitoring systems exist and could be offered to readers as additional examples. Salt Lake City's UUCON, Indianapolis INFLUX, UC Berkeley's BEACON. (Some of these networks may provide data to NOAA's GML data archive but their local/regional focus is worth noting.)

> We have revised the introduction to include these examples. Specifically, we cite examples from Indianapolis, Berkeley, and Salt Lake City, as suggested by the reviewer.

3. Approximate posterior covariance matrices, $\widetilde{\mathbf{Q}}_{\text{post}}$ and $\widetilde{\mathbf{\Gamma}}_{\text{post}}$ are given in line 264 and Equation B6 respectively. For the effectiveness of these approximation methods, the reader is referred to a citation: Saibaba et al., 2020. A few sentences within this manuscript describing results from Saibaba et al., the effectiveness of the approximations, limitations, etc. would benefit the curious reader.

> Thank you for this suggestion. We have provided additional details (lines 275-283) regarding the efficient representation of the posterior covariance matrix as a low-rank perturbation of the prior covariance matrix, and we have included some comments about accuracy of the posterior covariance matrix as well as the resulting approximate posterior distribution.

4. The two case studies reported in this work made use of pseudo-observations from NASA's OCO-2 instrument to estimate CO2 fluxes at 3-hour intervals and 1o x 1o spatial resolution. It is unclear from the text alone how OCO-2 soundings were incorporated into the inversion scheme. Generally, OCO-2 soundings are densely spaced ( 2-3km apart) and demonstrate varying spatial correlation in error. How is the assumption of spatially independent errors $\mathbf{R} = \sigma^2 \mathbf{I}$ justified?

> We have added additional detail to the methods section explaining these points. The STILT footprints from CarbonTracker-Lagrange are available every two seconds along the OCO-2 flight tracks. We use these footprints at two second intervals along the track to generate the synthetic observations used in the study. The spatial correlation in the observation and/or model error is less when using thinned observations at spaced intervals along the flight track. To our knowledge, most inverse modeling groups in the recent OCO-2 model inter-comparison study use a diagonal $\mathbf{R}$ covariance matrix, and we use that approach because it is prototypical of current inverse modeling studies (e.g., Peiro et al, 2022).

5. How/if soundings were spatiotemporally aggregated for this study is not clear. These details should be briefly included in the text while referring readers to other sources for more detail.

> The WRF-STILT footprints are available every two seconds along the OCO-2 flight tracks. We have added this information to the revised manuscript and refer the reader to Miller et al (2020) and Liu (2020) for additional information.

6. Figures 4 and 7 present the results of this work in a concise way; however, it is difficult to compare the effectiveness of so many different methods. Consider plotting the differences between the estimated and true fluxes. Obviously, the goal is to get as close to the true flux as possible. So, using a blue (-) to red (+) color gradient will easily show where the inversion is overestimating, underestimating, and effectively reproducing true emissions. Comparing gradients associated with relative differences may be easier across the various model outputs.

[Figure]

Figure 1: Difference images presented here correspond to results provided in Figure 4 in the manuscript. These images do not significantly improve interpretation of results, thus we continue to provide reconstructed fluxes, but remove one column of images for clarity.

Thank you for this comment. We agree that it is difficult to compare so many different methods, especially when we are showing time-averaged flux images for spatio-temporal reconstructions. As suggested, we looked at the difference in the average image (see Figure 1), but these images seemed even more difficult to digest. Thus, to aid the presentation of results, we decided to remove the column of reconstructions corresponding to the optimal regularization parameter. The optimal regularization parameter is not feasible in practice, and the results are very similar to the genHyBRs-dp and genHyBRmean-dp reconstructions (as evident in the relative reconstruction error plots in Figure 3.

**Technical Specifics**

1. Moving the citations in line 28 to immediately follow its associated observing system instead of listing them at the end of the statement would be helpful for readers.

   We have edited this line accordingly.

2. In line 82, it may be worth pointing out, for readers unfamiliar with this technique, that flux values from spatially explicit 2D arrays (x,y) are represented as a vector in this technique. Thus, n is the number of cells in the domain of interest. i.e. enforce how **s** is constructed.

   We have included clarification that **s** represents a vector containing spatial or spatio-temporal fluxes.

3. Minor point: In lines 52, 139, 150, 157, 247, 254, 265, 370, papers are referenced directly in the text but they still have parentheses around them. Typically, parentheses are only included if a statement is cited without direct reference in the text. (Is this GMD formatting?)

   Thanks for pointing this out. All in-text citations to references have been updated per GMD formatting instructions.

4. On line 247, the sentence beginning with "Instead, we follow the approach described in..." the word 'using' is included twice, making the sentence awkward to read.

> Fixed.

5. In line 291, p appears to be "filler" dimensions in the matrix to ensure that matrix multiplications work out. Although this can be intuited from the mathematics, it would be beneficial to point it out in the text.

> Yes, you are correct that this is purely for mathematical convenience, so that the problem can be reformulated for direct application of the genHyBR method. We have clarified this in the revision.

6. In line 343, $\sigma = 2$. It is suggested in the text that this corresponds to nlevel 100%. So, how can nlevel 50% be 0.5648 as in Table 1? Isn't the percentage of nlevel relative to this value from Miller et al., 2020? This is unclear in the text.

> Sorry for any confusion. In this paper, the choice of $\sigma = 2$ does not correspond to a noise level of 100%. In fact, $\sigma = 2$ corresponds to a noise level of 163% in the 6-weeks case study. We have clarified the definition of nlevel in the revision.

7. What are the units of $\theta_t$ and $\theta_g$ in line 370?

> $\theta_t$ has units of days and $\theta_g$ has units of km. We have clarified this in the revision as well.

8. Apparent typo in line 373: "... sparse can..." should be "... sparse and can..."

> Thanks for catching this. It is now fixed.

9. Figure #3 needs some work. The axes and title texts need to be made smaller to better fit the in- plot labels of $\lambda\sigma$. Smaller text will allow for bigger plot areas.

> Thank you for this suggestion. We have fixed the font size, removed one plot (corresponding to the direct method with $\lambda\sigma = 1$) and included a new plot containing the relative reconstruction errors per iteration of the LBFGS method (with fixed regularization parameter).

**2    Responses to Referee #2**

**General comments** The authors present new iterative computational methods for performing atmospheric inversions. They treat two cases, in which the mean of the fluxes is specified directly, and in which the mean of the fluxes is unknown and is described by a parametric linear model. Unknown covariance/regularisation parameters can be estimated as part of the iterative approach at relatively low additional computational cost. As a by-product of the method, an approximation of the posterior covariance matrix may be constructed. The methods are described clearly and the authors make a contribution to the state-of-the-art for atmospheric inversions. I support publication after the following concerns are addressed.

1. I think the authors could devote a bit more space to reviewing other iterative methods and placing their method in that context. For example, the method appears to fall within the broad class of variational methods. A few papers for these methods are cited but it would be useful for the reader to understand a little more about the proposed method in context, and (the subject of my next comment) how it improves upon existing methods.

> Thank you for this comment. Yes, our approach falls within a broader class of variational methods for solving inverse problems. In particular, a wide range of regularization terms can be included, and sophisticated optimization methods can be used to compute regularized solutions (corresponding to the MAP estimate). We have included a few references on variational regularization in Section 2.
>
> The most common variational method used in the atmospheric inverse modeling literature is L-BFGS. We have now comparisons to this approach (see next point). For estimating solution uncertainties, most researchers use Monte-Carlo simulations to approximate the uncertainties, although a a few groups use low-rank approximations. We now have results comparing our estimated uncertainties to other approaches.

2. Connected to the previous comment, I find it hard to evaluate the relative computational benefits of the method. It is clear that the method is faster than the direct inversion. However, is it faster (e.g., takes fewer iterations) than, say, methods proposed in Miller et al. (2020)? How does the computational complexity compare? Relatedly, all the methods tested make use of the fact that the transport matrix, H, was precomputed. That is no limitation of the method, which does not rely on this fact, but it does make it harder to understand the total time required for the inversion, which properly includes the precomputation of H. It would be good for the authors to mention this, and perhaps speak a little to the computational burden of calculating H. This is important because the authors suggest that one of primary contributions of the method is its computational efficiency.

> We have made various modifications to the results section to help with the evaluation and comparison of methods.
>
> - In Figure 3 of the revised manuscript, we removed the plot for the direct method with $\lambda\sigma = 1$, which we believe was not so informative, and replaced it with a plot of the relative reconstruction errors per iteration of the LBFGS method described in Miller et al. (2020). We remark that both the direct method and LBFGS require the regularization parameter to be fixed in advance, and a poor choice can yield poor reconstructions. Furthermore, although the the LBFGS approach considers mean estimation, this approach uses an improper prior for $\boldsymbol{\beta}$ rather than a Gaussian prior.
>
> - We observe that the generalized hybrid methods converge faster than the LBFGS method, which is significant because the main computational cost per iteration (i.e., one matrix-vector multiplication with $\mathbf{H}$ and its adjoint) is the same. In fact, running the forward and adjoint model is often the most intensive step of the inverse model, and a goal of this paper is to devise methods that decrease the overall number of iterations required (i.e., a smaller number of iterations means less computing time).
>
> Furthermore, GenHyBR has the added advantage that it simultaneously estimates good regularization parameters during the solution process and estimates uncertainty at little additional cost (in particular, no addition matrix-vector products with $\mathbf{H}$. This makes GenHyBr much more favorable especially when actual transport models are used.
>
> The actual time required by the forward and adjoint models depends on the model, which gas is being modeled, and what spatial resolution is being used. The forward and adjoint models used in

this paper were generated by our collaborators at NOAA, so we do not have exact timings. However, we emphasize that the genHyBR algorithms can be paired with models where matrices do not need to be computed explicitly.

3. The authors present uncertainty quantification for the fluxes but, unlike the reconstructed fluxes, there is no evaluation of whether the approximate covariance matrix is suitable. My concern would be that the low rank approximation could over estimate marginal variances and underestimate correlations (though whether this is true is not obvious to me). A second concern is that the regularisation parameter $\lambda$ is changing in every iteration (probably more so in earlier iterations), whereas I believe the theoretical results for the posterior covariance in Saibaba et al. (2020) depend on knowing $\lambda$ (though I may have misunderstood this). These concerns could be addressed empirically by evaluating the suitability of the uncertainties. One simplistic way do this would be to investigate this would be to compare the approximate posterior standard deviations to those from the direct method, which should be available for the six week case study. A second way would be to discuss in slightly more detail what the results of Saibaba et al. (2020) say about this. A third way, perhaps less feasible, would be to consider a scoring rule that takes into account the uncertainty (similar to the reconstruction error for the posterior mean). One such scoring rule could be the posterior log density (which is multivariate Gaussian) evaluated at the truth—higher density would be better. Another scoring rule option could be to consider the calibration of the uncertainty intervals for some quantity of interest such as the total flux (for example, do the 50% prediction intervals for the total flux in each time step contain the total flux 50% of the time?) These ideas are just meant as suggestions, the more general request being to explore whether the uncertainties are suitable.

Thank you for your ideas and suggestions. The uncertainty estimates in this paper are overestimates rather than underestimates. This is because the covariance approximation is of the form

$$\lambda^{-2}\mathbf{Q} - \mathbf{Z}_k\boldsymbol{\Delta}_k\mathbf{Z}_k^\top,$$

that is a negative definite update of the prior covariance matrix. As the number of iterations increase (and $\lambda$ is fixed; see discussion below) the uncertainty decreases monotonically. It is likely to be better in overestimating rather than underestimating uncertainty. Regarding the choice of $\lambda$, we follow the approach described in Saibaba et. al (2020), where the genHyBR method is used to estimate the regularization parameter and build up a good subspace. Estimating the regularization parameter on-the-fly is actually a feature (rather than a concern) of a hybrid approach (such as ours) which allows us to determine a good stopping criterion and ensures that the number of iterations remain small overall. Then, for subsequent UQ, the regularization parameter is assumed to be fixed, and all following results depend on this choice.

We incorporate some of this discussion in the revised manuscript (Section 3.2).

**Specific and technical comments**

1. In Section 3.1.2, it would be good for the authors to provide a reference that specifically discusses the use of the discrepancy principle in similar problems.

Although widely used in the inverse problems and imaging science communities, there are only a few studies in atmospheric inverse modeling that have used the discrepancy principle as a regularization parameter selection method. For example, in Hase et al (2017), the DP method was used to estimate regularization parameters for a sequence of Tikhonov-regularized problems for sparse reconstruction. However, a major distinction here is that by using the iterative hybrid formulation, we can exploit relationships from the projected problem for more efficient parameter estimation for DP (c.f., equation (16) in the revised manuscript) as well as for other parameter estimation methods (c.f., Appendix A).

> In the revised manuscript, we have included the reference above in Section 3.1.2, as well as additional references of the discrepancy principle used in other inverse problems applications.

2. On page 14, line 358, it is stated that X is identical to Miller et al. (2020). But in Miller et al., X is described as having only one column, an intercept. Have I missed something?

> This is a good point. In Miller et. al., it should be clarified that "the matrix $\mathbf{X}$ only consists of columns of ones" and zeroes. In the revision, we have provided a mathematical formula for construction of $\mathbf{X}$ as $\mathbf{X} = \mathbf{I}_8 \otimes \mathbf{1}$ and additional details.

3. At several places in-text citations are surrounded in parentheses, e.g. written as (Smith et al., 2020) instead of Smith et al. (2020). Generally I would expect to see the latter when the citation is incorporated into the sentence.

> Thank you for pointing this out. All in-text citations to references have been updated per GMD formatting instructions.